# Finding help and hope in a peer-led reentry service hub near a detention centre: A process evaluation

Arthur McLuhan[1], Tara Hahmann[1], Cilia Mejia-Lancheros[1], Sarah Hamilton-Wright[1], Guido Tacchini[1], Flora I. Matheson[1,2,3]*

1 MAP Centre for Urban Health Solutions, St Michael's Hospital, Toronto, Ontario, Canada, 2 Dalla Lana School of Public Health, University of Toronto, Toronto, Ontario, Canada, 3 Centre for Criminology and Sociolegal Studies, University of Toronto, Toronto, Ontario, Canada

* flora.matheson@utoronto.ca

**Data Availability Statement:** This study reports on qualitative data which by nature is difficult to fully anonymize. Participants were not asked to agree to

## Abstract

When people leave correctional institutions, they face myriad personal, social and structural barriers to reentry, including significant challenges with mental health, substance use, and homelessness. However, there are few reentry programs designed to support people's health, wellbeing, and social integration, and there are even fewer evaluations of such programs. The purpose of this article is to report the qualitative findings from an early process evaluation of the Reintegration Centre—a peer-led service hub designed to support men on the day they are released from custody. We conducted semi-structured qualitative interviews and examined quantitative service intake data with 21 men who accessed the Reintegration Centre immediately upon release. Participants encountered significant reentry challenges and barriers to service access and utilization. The data suggest that the peer-led service hub model enhanced the service encounter experience and efficiently and effectively addressed reentry needs through the provision of basic supports and individualized service referrals. Notably, the Reintegration Centre's proximity to the detention centre facilitated rapid access to essential services upon release, and the peer-support workers affirmed client autonomy and moral worth in the service encounter, fostering mutual respect and trust. Locating reentry programs near bail courts and detention centres may reduce barriers to service access. A peer-led service hub that provides immediate support for basic needs along with individualized service referrals is a promising approach to reentry programs that aim to support post-release health, wellbeing, and social integration. A social system that fosters cross-sectoral collaboration and continuity of care through innovative funding initiatives is vital to the effectiveness and sustainability of such reentry programs.

## Introduction

More than 30 million people experience incarceration annually worldwide, with nearly 11 million people held in custody on any given day [1]. Although nearly all incarcerated individuals will be released, most are rearrested and about half are reincarcerated within three years [2,3].

their data being available for other studies, as they experience significant social and health burdens that are highly stigmatized, including criminalization, homelessness, and poverty. Requesting that they agree to share their data more publicly would have deterred some from taking part in the study and would have the potential to significantly alter the data obtained. Ethical approval for this research study was granted by St. Michael's Hospital Research Ethics Board [REB 17-077] on the basis that participants' data was only accessible by the research team. The authors believe in the principle of making data freely available but this was not anticipated or specifically approved by the St. Michael's Hospital Research Ethics Board at the time the study was initiated. However, specific requests received by the authors for data sharing for purposes such as data verification and meta-analysis will be considered by the St. Michael's Hospital Research Ethics Board on an individual basis under defined and mutually agreed-upon conditions.

**Funding:** FM received project funding from the Ontario Government and the Ontario Trillium Foundation [LP95856]. The funders had no role in study design, data collection and analysis, decision to publish, or preparation of the manuscript. https://otf.ca/; https://www.ontario.ca/

**Competing interests:** The authors have declared no financial or non-financial competing interests exist.

The consequences and costs of failing to break the cycle of incarceration are not contained to incarcerated populations—they spillover into and reverberate through families, neighbourhoods, communities, as well as justice, health, and economic systems [4–7].

Community reentry has become an important policy, practice, and research concern [8], and reentry programs are often positioned as important tools in the effort to improve post-release outcomes [9]. Although the reentry movement in corrections has tended to emphasize recidivism-focused program objectives, outputs, and outcomes, there has been increasing recognition of reentry concerns beyond recidivism [10,11].

When people leave correctional institutions, they contend with a variety of reentry challenges. Low levels of formal training and education [12–14], criminal records [15], limited employment opportunities [16–21] and other intersecting factors, including racism and discrimination [22], detail some of the complex socio-economic reentry barriers facing this population [23–25]. Secure housing is often the principal immediate need upon release from custody [4,8,26,27], but poverty and a lack of affordable housing contribute to high rates of homelessness and housing instability. Compounding housing access barriers and, in turn, reentry challenges [28–31] are serious health concerns [16,26,32–34] and inadequate treatment and care [19], which contribute to post-release mortality and overdose [35–37], the highest risk occuring in the immediate week and month following release [37]. Releasees also have high rates of mental illness, substance use, trauma, traumatic brain injury, and infectious disease [38].

Whereas quantitative research studies have identified the prevalence and severity of some of these challenges, qualitative studies [39–41] demonstrate how people experience and give meaning to reentry as a social process, and are thus well-positioned to offer invaluable insight into the interpretive and relational dimensions of reentry and the challenges thereof [41]. For example, qualitative research on "prisonization" shows the subtle yet significant ways incarceration can shape patterns of thinking, feeling, and acting, and that these dispositional orientations can have enduring effects in the lives of persons with experiences of incarceration, spilling over into the sphere of community reentry and making social integration more difficult [42,43].

Many individuals leaving correctional facilities are also subject to community supervision orders. Qualitative research on the "pains of probation and parole" [44–49] reveals the interactional work involved in managing the practical challenges of complying with restrictive supervision conditions while pursuing reentry and reintegration. How individuals define and respond to supervision conditions is a "contextually contingent" [50] and "interactive" [51] process, "depending upon their previous experiences, background, social context, and expectations about the penalty" [50: p.382]. Differences in how conditions are set, monitored, and enforced can reinforce inequities—such as those tied to race [52], gender [53,54], and health [55]—which are further amplified by underfunding and increasing cuts to programming for mental health issues, substance use, and other transitionary planning [40].

More generally, qualitative research demonstrates the relationship between social identities and reentry obstacles—e.g., the mark of a criminal record [22,56]—and opportunities—e.g., the hope for future selves and situations [57]. Thus, while reentering individuals are often focused on "moving forward" and "making good" [39], like the desistance process [58], reentry trajectories are non-linear. Encountering obstacles and experiencing setbacks are common challenges and feelings of embarrassment, shame, and frustration that result from reentry challenges can erode commitments to positive change [22,39,40,56,57,59]. There is increasing recognition that such challenges require approaches to reentry programming with more comprehensive and compassionate understandings of reentry contexts and care than recidivism-focused models.

Among the alternatives to recidivism-focused reentry programming are those that seek to support health and social integration [5,25,60]. The complex health challenges of incarcerated populations upon release require "individual, client-centered discharge planning, especially since so many returning prisoners reported multiple types of health conditions. The actual services provided to individual returning prisoners must be targeted to their particular needs" [60: p.5]. However, the neglect of discharge planning in many jurisdictions coupled with the range of service barriers that mark "the steeplechase of reentry" often means that needs go unmet during the immediate days and weeks of the post-release period, which are critical to making a successful transition from custody to community [61] and mitigating some of the worst post-release risks and outcomes [35].

There are some models of reentry programming that focus on connecting people to targeted services immediately following release. For example, to help ease the transition from the institutional sector, cost-effective coordinated solutions such as integrated service hub models of care can help facilitate collaboration and resource sharing to improve access to community care [62]. Service hubs support reentry by providing one setting in which clients can connect to a continuum of care [63–67]. They are particularly important when the clientele have complex social and health needs, including mental health and substance use [68]. The community partnerships at the heart of hub models not only prepare clients for community reentry, but also prepare local communities to better support individuals returning to the community [69].

Such transitional service models may include additional program components and features found in models of reentry programming that emphasize goals and measures beyond recidivism. For instance, although many programs make efforts to identify and address clients' reentry needs and risks, part of the paradigm shift is a move from deficits-based to strengths-based orientations to reentry, emphasizing and empowering the competence, autonomy, and agency of people with experiences of incarceration [70,71]. Similarly, peer support has also been incorporated as a component of reentry programs for both those providing [57,72] and accessing services [73]. For example, some evidence indicates that peer support workers who work in programs that include a strengths-based approach to service delivery and client support experience positive self-identity transformations in which they begin to experience feelings of "giving back" and redefine themselves as being part of the solution rather than the problem [61]. Such redefinitions of self and situation have been associated with not only an improved ability to maintain their own desistance [57], but also a variety of benefits for their clients reentering society [39,74], such as supporting access to treatment, enrolment in training and education, work and housing placements, as well as connections to other supports and resources [75]. In the context of peer recovery support for the previously incarcerated with substance use disorder, improvements were found in self-rated mental and physical health and a reduction in substance use in one study [73]. A systematic review found similar improvements to substance use and other recovery outcomes among those participating in the peer support recovery services [76]. These findings suggest that peer support workers are uniquely positioned to support re-entering individuals through shared experience, understanding, and respect [77].

In contrast, justice-involved individuals have expressed concern over the ability of community corrections officers (CCOs) to support them given a lack of shared experience and understanding. Two separate studies found that social distance and doubt around the sincerity in CCOs' efforts to help them reintegrate are barriers to successful reentry [30,78]. While adequate training may reduce some barriers, especially training that helps CCOs better understand the post-release needs of their clients [78], peer support workers offer a unique capacity to support one another through shared experience. An additional challenge for CCOs in supporting clients is the decentralization of services, which increases the time and effort required

to make referrals to appropriate supports, leaving those seeking reentry with delayed connections to necessary services and supports [79]. Some holistic programs have sought to meet complex reentry needs with more collaborative programming that includes peer support. For example, the IF project based out of Seattle, Washington offers a volunteer reentry mentoring program where mentees and mentors are matched pre-release and continue their mentorship post-release embedded in a larger collaborative structure of individuals with lived experience, law enforcement and community partners [80].

Evaluation research can inform the design and delivery of transitional reentry programs, but there have been comparatively few evaluations of transitional reentry programs focusing on health and social integration relative to recidivism-focused models [81]. A notable exception is the Kinner, Lennox [81] evaluation of a health-focused transitional service brokerage intervention for people recently released from custody, which demonstrated the utility and cost effectiveness of harnessing existing community services to meet the specific needs of clients.

In this article, we report the core findings of a process evaluation [82] of a reentry service hub designed to offer housing and social support services and referrals to men released from the Toronto South Detention Centre, the largest detention centre in Ontario, Canada. Process evaluations determine whether program activities have been implemented as intended and resulted in certain outputs. Process evaluations can answer questions such as, "How well the program is working. The extent to which the program is being implemented as designed. Whether the program is accessible an (sic) acceptable to its target population" [83: p2]. Process evaluations are particularly well suited to assessing the implementation of interventions that address complex challenges and contexts such as reentry. A small number of process evaluations have been conducted on reentry programs in the United States using qualitative data to report on the lived experience of those navigating the reentry process upon release and/or understandings of program success among attendees [84,85]. Often process evaluations proceed with qualitative methods to assess client experiences with the intervention [86] and may provide additional qualitative details (e.g., details on staffing and program delivery) to help build contextual insights [87].

We drew on qualitative interviews with Reintegration Centre clients to assess their experiences related to the challenges of reentry, services of the Reintegration Centre, and the role of peer support workers. In what follows, we first describe the study setting and methods. We then report the main qualitative findings of the process evaluation. We conclude by discussing the findings in the context of reentry practice, policy and research.

## Method

This study is part of a larger participatory research [88], mixed-methods process evaluation [82] of the Reintegration Centre, a reentry service hub for men released from custody. We worked collaboratively with our community partner John Howard Society of Toronto throughout the study, from design to dissemination. Consistent with participatory action research methods [89], the research team conducted the analysis for the evaluation independently from the partner agency staff, but we worked together to develop recommendations from the findings. The study followed the principles of the Canadian Tri-Council Policy Statement: Ethical Conduct for Research Involving Humans [90]. The Research Ethics Board [REB # 17–058] of St. Michael's Hospital, Toronto, Ontario, Canada reviewed and approved the project. All participants received honoraria.

## The reintegration centre

At the time of the study, the Reintegration Centre was located in the largest metropolitan area in Canada within walking distance to Toronto South Detention Centre, a province-operated, maximum-security facility for adult men and transgender people of different genders serving sentences less than two years or being held on remand and awaiting to appear before the courts. The environment around the Toronto South Detention Centre is a service desert and considerable distance from the city of Toronto, with the Reintegration Centre strategically positioned nearby to ensure immediate access to practical supports. The John Howard Society of Toronto (JHST)—the organization that established the Reintegration Centre—is a service organization committed to supporting people who have been affected by the criminal justice system. The Reintegration Centre was designed to support clients upon release from custody, a critical point in the reentry journey that can impact successful reintegration [91]. Staffed by peer support workers with lived expertise of criminal justice involvement, the Reintegration Centre provided clients with immediate supports, including access to clothing, food (snacks and gift cards), transportation (public transit tokens), and harm reduction resources, including overdose prevention education and Naloxone kits. Based on client needs and priorities, peer support workers also provided individualized referrals to a continuum of supports within JHST (e.g., housing, addiction and harm reduction, and domestic violence services) and other service agencies, including shelter, mental health, social assistance, and employment service providers.

## Procedure

JHST employed professionally trained peer support workers at the Reintegration Centre. These individuals had lived experience of incarceration and substance use. Ample research shows that there is incredible value to lived experience/peer support for both peers and clients. Peer support workers engage clients to offer hope, support, encouragement, and an intimate understanding of the barriers and issues that they are facing because they have been in a similar situation [92].

## Recruitment

Peer support workers introduced the study to each client after a standard intake needs assessment interview, during which they asked about client needs and strengths. Staff introduced the study to 209 clients, and 139 consented to be part of a study and to share their intake needs assessment data with the research team, of which 49 also provided informed written consent to participate in a qualitative interview with the research team and 21 completed the interview. Of the remaining 28 participants who consented to be contacted but did not complete the interview, we attempted re-contact but could not reach them via telephone or e-mail. As others have noted, the challenges of reentry may also pose challenges to research participation and retention [93]. Participants received honoraria as follows: $10 at study enrolment, $5 when an interview date was scheduled, and $30 plus two transit tokens for the interview. Participants were enrolled between September 19, 2017 and December 21, 2018. The average length of follow-up was 27 days (range = 18–43).

## Data collection and instruments

Quantitative data to describe the study sample was collected during the intake needs assessment (S1 File) between the client and the peer support worker. This needs assessment is used at the Reintegration Centre to identify immediate needs of the clients and to set priorities for

support in a short interview format. It includes questions on criminal history, ratings of health [94,95], parental roles, history of trauma (the latter adapted from the Brief Trauma Questionnaire) [96], resiliency [97], gender and ethnic identity, social support, service access, head injury, immediate and priority needs, and client strengths. This needs assessment provides a snapshot of the key health and social needs the participants faced at immediate release from custody, provides insight into the context of program delivery, and aids in the interpretation of qualitative findings from the process evaluation. The intake assessment was developed by/ with the Centre for Addiction and Mental Health (CAMH: Toronto, Ontario) and JHST—and prior to the development of the research project—with several goals, including peer and client empowerment through questions that centre on client well-being and strengths. CAMH staff provided the peer support workers with training that would support the peers to complete the intake form using the principles of Motivational Interviewing, empowering the peer support workers with the freedom to complete the tool in a conversational manner. In contrast to the deficit-based models that dominate the field, the form is a 2-way empowerment tool for peer support workers and clients.

Reintegration Centre staff and researchers co-developed a semi-structured qualitative interview guide (S2 File). The interview guide was designed to provide an understanding of service delivery experiences through the Reintegration Centre including peer support, broader experiences of reentry/reintegration, and priority needs at release. Included in these broader areas of interest are topics such as resiliency, housing, health and illness, sources of personal support, coping strategies, family and parenting, relapse and harm reduction, finances and employment, education/training, and personal goals for the future. Interviews ranged from 18 to 172 minutes, with an average of 49 minutes in length. Interviews were audio recorded on an encrypted device and transcribed verbatim. To ensure confidentiality, we assigned each participant a study identification number on all study materials. In the spirit of participatory research, the research team developed and implemented training for Reintegration Centre staff who would help with recruitment and for the peer support workers who would conduct the qualitative interviews. As part of this training, experienced members of the research team shadowed the peers during early interviews to ensure consistency of questioning, to enhance interviewing skills, and to debrief with peers after interviews.

## Participants

Participant ages ranged from 23 to 53 years, with a mean of 37 years (Table 1). Most participants faced material precarity and hardship on release, with 71% identifying housing as a priority need.

## Data analysis

Consistent with the analytical approach of other qualitative process evaluations [84,85,98], we conducted open coding of the qualitative data derived from participant interviews with a group of 5 coders. Coding began with 4 transcripts. Each coder [GT, DK, NJ, PD, AB] was assigned 2 transcripts so that each transcript was reviewed by at least 2 people. We selected transcripts to reflect different points in the data collection process. At this first stage of the coding process, each coder developed preliminary codes, and then shared and discussed them with the rest of the team. Each team member was then assigned two additional transcripts to review for high-level codes within the data. The team met to review the codes and develop code names, definitions, and coding rules to ensure consistency among coders. This information formed the contents of the codebook.

Table 1. Descriptive statistics for the study sample (n = 21).

| Variable[a] | Mean (n) |
|---|---|
| Age | 37 (21) |
| Months in custody | 1.80 (20) |
| | % (n)[b] |
| **Health status indicators**[c] | |
| Self-reported physical health (excellent/very good) | 47.2 (*9*) |
| Self-reported mental health (excellent/very good) | 47.4 (*9*) |
| History of head or neck injury | 71.4 (*15*) |
| **Socio-demographic status indicators** | |
| Has family doctor | 45.0 (*9*) |
| Has children | 42.1 (*8*) |
| **Correctional status indicators** | |
| Sentenced vs. remanded | 68.4 (*13*) |
| Recently in custody for a new charge | 47.6 (*10*) |
| First time in custody | 23.8 (*5*) |
| **Traumatic life event indicators** | |
| In a serious accident | 47.6 (*10*) |
| Feared serious injury/being killed | 61.9 (*13*) |
| Witnessed or feared serious injury or accident | 61.9 (*13*) |
| Self-reported resiliency (Strongly agree/agree) | 84.2 (*16*) |

[a]Not all response categories shown.

[b]Percentages may not add to 100% due to missing values.

[c]Not discussed was a response category for these indicators: Head injury, family doctor, children, sentenced, new charge, and traumatic event indicators.

To pilot the codebook, we conducted focused coding where team members (as noted above) independently coded 2 additional transcripts. The team [GT, SHW, PD, AB, NJ, FIM] met to discuss coding progress and consistency. If new codes emerged, they were integrated into the codebook. The full data set of 21 interviews was coded [GT, SHW, PD, AB]. We identified 32 codes in total.

We focused on two particular codes "Personal Experience of Reentry" and "Reintegration Centre Services" as well as the responses to two specific questions, one on post-release priorities, and another on service referrals. We performed an inductive qualitative content analysis [99] of these data to assess Reintegration Centre clients' personal experiences of release, reentry, and reintegration. One coder [AM] focused on the themes that addressed personal experiences of release, reentry, and reintegration; another coder [CML] focused on the themes that addressed experiences of Reintegration Centre services; and two coders [AM and CML] analyzed responses related to post-release priorities and service referrals.

## Results

### Description of study sample

The men in this sample experienced multiple health issues and navigated a complex social landscape (see Table 2). Participant contacts and conflicts with the criminal justice system were diverse, varying in terms of in-custody periods, reasons for custody, health status and traumatic life events. Participants spent an average 1.8 months in custody, reflecting the high numbers of men on remand. For one quarter (24%) of the sample, this was their first time in

**Table 2. Detailed sample characteristics on complexity (n = 21).**

| ID | Age | Recent Incarceration | Type of custody | First custody | Good physical health | Good Mental health | Family Doctor | Children | Head, neck injury | Serious accident | Faced environmental, ecological disaster | Faced, feared serious injury, being killed | Witnessed, feared serious injury, being killed | Bounce back quickly after hard times |
|---|---|---|---|---|---|---|---|---|---|---|---|---|---|---|
| P01 | 46 | . | . | | x | x | | | | x | | x | | x |
| P02 | 23 | New charge | Sentenced | | | | x | | | x | x | x | x | x |
| P03 | 26 | New charge | Sentenced | | | x | | x | | | | | | x |
| P04 | 30 | New charge | Sentenced | | | | | | | | | | | x |
| P05 | 49 | Breach | Sentenced | | | | x | x | x | x | | x | x | x |
| P06 | 23 | Breach | Sentenced | | | x | x | | x | | | | x | x |
| P07 | 24 | ~ | Remand | | x | x | | x | x | | | | | x |
| P08 | 47 | New charge | Sentenced | | x | x | x | x | x | x | | x | x | x |
| P09 | 48 | Breach | Sentenced | x | | | | | x | x | x | x | x | |
| P10 | 24 | New charge | Remand | x | x | x | x | | x | | | | x | |
| P11 | 26 | Breach | Sentenced | | | | | x | x | | | x | x | . |
| P12 | 27 | New charge | Sentenced | | x | x | | | x | | | x | x | x |
| P13 | 51 | . | Remand | x | | | x | . | x | | | x | | x |
| P14 | 52 | Breach | . | | | | x | x | x | x | | x | x | x |
| P15 | 25 | Breach | ~ | x | x | | | | x | | | x | x | |
| P16 | 42 | Breach | Sentenced | | x | x | | x | x | x | x | x | x | x |
| P17 | 52 | ~ | Remand | | | | ~ | | x | x | | x | x | x |
| P18 | 36 | New charge | Remand | | . | . | . | . | x | x | | x | x | . |
| P19 | 24 | New charge | Sentenced | | x | x | x | | | | | | | x |
| P20 | 52 | New charge | Sentenced | | x | | x | x | x | x | | | | x |
| P21 | 53 | New charge | Sentenced | x | . | . | ~ | ~ | ~ | ~ | ~ | ~ | ~ | x |

custody. Nearly half of the participants (48%) were in custody for a new charge, two-thirds (68%) were serving sentences while 26% were on remand.

Although nearly half of the participants (47%) reported excellent or very good physical or mental health, most participants also reported experiencing complex health issues and social needs. Nearly three quarters of participants (71%) reported a head or neck injury. About half (48%) had been in a serious accident, nearly two thirds had experienced or feared serious injury or being killed (62%) or witnessed (or feared) a situation in which someone was seriously injured or killed (62%). Most participants faced material precarity and hardship on release, with 71% identifying housing as a priority need. While many experienced multiple social and health challenges, most participants (84%) reported that they tended to bounce back quickly following hard times.

In what follows we set the stage by outlining the challenges people experiencing community reentry face. We then consider participant experiences of the Reintegration Centre across three main themes: (1) providing immediate supports, (2) making referrals, and (3) becoming a trusted reentry resource.

## The challenges of community reentry

The Reintegration Centre aimed to facilitate the transition from incarceration to community. Before considering the specific features, functions, and meanings of the Reintegration Centre as a reentry resource, we overview the personal, social, and structural context of community reentry for study participants revealing that it is beset with challenges.

When we asked participants whether they felt ready to be released, most expressed that one is always ready to leave jail. For example: *"No one wants to be in there. It's terrible"* (P12) and, *"Any day in jail is a long time"* (P17). However, wanting to get out and being prepared to make it on the outside are different matters. Short detention periods coupled with rapid service access, supportive relationships, gainful employment, and stable housing upon release facilitated community reentry. For example:

> *It was my first time doing time, and it was only 28 days, so it [reentry] wasn't a huge adjustment. It's not like I was in a foreign city or a faraway city for years or something, so I could basically just pick up where I had left off... Where I was living before I went in and where I went after is the same place. I have my common-law wife I've been living with for four and a half years, so that relationship is solid, and I didn't have any trouble because of housing due to that situation ... My parents are supportive. They helped me pay rent for the month of August so that was good, to have that family support. They paid rent and an extra $500, so that's helped my wife and I keep our heads above water ... I had the job interview with [a delivery service] and I passed the interview, I passed the online computer test, my driver's abstract is clean so that was a positive thing that makes me feel upbeat. I'm also applying for other jobs. I applied for four jobs today, so I'm sure I'll find something good soon ... I have a fair bit of university education.* (P21)

While the supports for successful reentry were available to some, most participants lacked them. As one participant put it, "*I didn't really feel ready [to return to the community] ... because I had nothing to go back to*" (P10). Housing was hard to find, employment opportunities were limited, income supports were inadequate, relationships were strained or severed, and accessing needed services was often difficult and delayed:

> *I felt unprepared. I didn't feel really prepared at all ... [T]hey release you, and they just let you get back into the community ... [Then] it was a six-week wait just to see a social worker.* (P11)

> *[I am] kind of like in the middle of nowhere right now just lost my place, lost, I haven't had a steady job in a couple of years. I keep getting into trouble with the law and things could get a lot better. I'm kind of like at ground zero right now when it comes to being at a place in my life. I'm kind of distant from family and relatives.* (P04)

In the context of those challenges, the desire to leave detention may be tempered by reservations about returning to the community. Reentry reservations are best understood in social-structural and comparative context. That is, for people with experiences of poverty, homelessness, isolation, discrimination, stigma, and a variety of other struggles in society, being incarcerated may offer relative stability and security over returning to the community:

> *[M]ost people will say 'Oh I was scared to be in jail. I was scared to get put in.' I was scared to come out because when I went in I was already in a homeless situation. I was already wondering when my next meal was going to be. When I got put in jail, you know, I was getting clean, I was getting healthy, I was putting on weight, I was eating three square meals a day, I was getting showers, [and] I was in a warm place. I knew that I'm in this concrete box, I'm not going anywhere, [and] I'm warm. What scared me [about reentry] was: Am I going to relapse when I get out? Am I going to fuck this up?* (P07)

Thus while participants were eager to leave the confines of detention, they also worried about the uncertainty, insecurity, and adversity they would face in returning to the community with few resources and supports.

Together these challenges would make breaking the cycle of incarceration difficult. Given the reentry obstacles encountered upon release, some participants perceived a lack of viable options beyond a return to earlier habits, routines, relationships, and ways of life that risked further criminal justice involvement:

> *[The challenge has been] trying to keep my nose clean. It's hard. It's like I'm having trouble finding a job and shit, having trouble with ID. And it's like outside [i.e., on the street] it's easier for me. It's easier for me to like, you know, do other [less than fully legal] things and get money, but it's like I don't want to. . .* (P12)

> *There's no options when we come out of jail, so the [post-release] reality is why the jail is so packed . . . it's kind of like a catch 22.* (P05)

Once you have been in the criminal justice system, you are more likely to return. Still, most participants hoped to defy the odds and avoid further conflicts with the law, and they identified reentry support as an essential tool in that effort. For example, in reflecting on his reentry experience, including the support he received from the Reintegration Centre, one participant bemoaned the lack of discharge planning support inmates receive while incarcerated:

> *They [the correctional system] don't prepare us to help us out [in returning to the community], like they should be doing stuff like this [the work of the Reintegration Centre] while we're in there . . . so things are already set for [us], instead of us coming out like chicken [with their] heads [cut off] and running around.* (P12)

In sum, participants confronted a variety of personal, social, and structural reentry challenges. With histories marked by homelessness, addiction, violence, trauma, discrimination, and so on, their experiences of struggle in society made them well aware of the challenges that lay ahead. Especially telling were the reservations they experienced as they approached release. All wanted to leave, but many were afraid to go. Some men felt more stability and security inside than in the broader community. They were willing to challenge their ostensible fate, but noted they would require targeted support along the way if they were to have a fighting chance.

## Providing immediate supports

The Reintegration Centre was often participants' first post-release service interaction—*"Well, the first day I got out . . . I came to see you [peer interviewer/support worker at the Reintegration Centre]."* (P07)—and the Reintegration Centre's proximity to the detention centre facilitated these first contacts, providing rapid access to services:

> *I like this resource centre. It's kind of actually one of the better ones and it's probably because it's right next to the jail. . . and it's nice that you guys have a place like this right from when you're released.* (P04)

> *I'm really glad [the Reintegration Centre] was there the day I got out. I think it's wonderful to have a location by the detention centre. It made a difficult day a little less difficult when I got out, and just I'm glad you're around. I think it's really important work.* (P21)

At intake, Reintegration Centre peer support workers asked participants to identify up to 3 priorities they would be focusing on as they returned to the community. Participants identified 17 types of priorities. Housing (n = 14) was the most common priority. A cluster of other priorities followed housing: drug treatment (n = 5), clothing (n = 4), transportation (n = 4), employment (n = 4), education (n = 4), government identification cards (n = 4), financial assistance (n = 4), and relationships (n = 4).

The Reintegration Centre met some of these priorities on site, such as immediate clothing, food (snacks and gift cards), and transportation needs (public transit tokens). One participant noted that the transit token was his only priority that day, and another commented, *"I really needed it [transit token] that day"* (P06). The need for suitable clothing was particularly acute in the winter, and participants were appreciative of the service: *"[The Reintegration Centre provided] the [winter] jacket. It was a very nice jacket"* (P06). Similarly, a number of participants found the gift cards helpful in addressing some of their pressing food needs: *"I'm used to going without food . . . the [food gift] card you guys [provided is] help[ing] me survive"* (P08).

All participants were offered harm reduction services, which are critical in the context of the ongoing opioid epidemic and the proliferation of fentanyl, fentanyl analogs, and novel synthetic opioids in illicit drug markets. Some were not involved in drug use and expressed no need for these services. Others declined the services because they only used "soft drugs" (e.g., marijuana). Still others declined the service because they felt they were already equipped with the necessary harm reduction knowledge and materials for their situation, as in *"I know the protections and that stuff"* (P02) or *"I had my own [harm reduction] kit already"* (P01). Participants who accepted the harm reduction services found them to be essential. For example, when asked, "Was the information helpful that the Reintegration Centre gave you about harm reduction?" one participant responded:

> *Yeah, absolutely, on how to use, how to use safely and all that stuff, yeah, you know. And also, they made me aware that there is safe injection sites. I don't have to use on the street, which was another good thing that they offered.* (P17)

Together, these first contacts and immediate supports not only provided basic sustenance and security but also contributed to an overall sense of well-being in the first moments post-release. As one participant told us, *"those things are huge to keep your composure, and I had nowhere else to go, so it was very important"* (P08) and another said the Reintegration Centre encounter set the stage for an encouraging release day, *"You guys helped me out. It was a good day, actually"* (P06).

Interactions with social service providers can often be frustrating, even humiliating, especially for people experiencing complex needs [100]. For example, when asked if there were times "when you reached out or needed help and you didn't get it?" one participant lamented, *"All the time. That's regular. Nobody wants to help. Most people just want to act like. . . 'Sit there and listen'—and then tell you you're the problem, everything is your fault!"* (P12). Others expressed similar frustrations. For instance,

> *Well the [reentry] challenges are just trying to get everything done and nobody's helping me so I just, like I said, I just put my hat, panhandle and make my money to get by . . . I'm trying to get my income together and everything else just screwing me up . . . You know, I make an appointment . . . for the clothing bank and the old lady was a bitch with me on the phone . . . She wasn't paying attention. Oh, this is, rah, rah, rah, rah so yeah. I'm stressed out . . . [the disability support program] won't give me any money. Welfare won't give me any money. I have an appointment coming up [with another service provider] for housing, so if I don't have*

*the income, I got no money. I got no income and they can't do anything for me so I'd lose my subsidized housing. It's just been rough [returning to the community] . . . All I do is put out my hat every day and panhandle and so I have enough money to live off of . . . and also running around trying to get my [disability support application] together, trying to do Welfare, trying to do whatever I can getting it together so I can get a life.* (P13)

Similarly, after being informed that he might be in violation of his probation order—almost immediately upon release—for accessing the services of the Hub to arrange access to drug treatment, rather than reporting directly to a drug treatment program, another participant shared:

*[I was told] 'You might get charged for a breach, you know.' I had enough, like when you have that pressure, it makes you want to go use . . . These people are . . . not even helping you and probation is supposed to help people, like you're on probation, but they're supposed to tell you, guide you, help you, and get you help. It seems like they don't do that.* (P05)

People who need support the most often go without, avoiding anticipated difficulties with providers. When they pursue service access in spite of their reservations, the encounter is transactional: go in, apply, grin and bear whatever challenges arise, and get out. In contrast, participants told us that the Reintegration Centre is much more than an impersonal conduit to reentry resources. The Reintegration Centre encounter left them with the sense that their lives mattered and that their dignity and rights as persons were affirmed:

*When I went and I got out and I talked to the person [at the Reintegration Centre], she cared. It wasn't more like paperwork. It was more like you know, 'Sorry to hear that.' It was a human and I felt human. I didn't feel like a guy who came out of jail . . . I talked to her about what's going on in my day, what's going on with me so there's still that [sense] you're a human being and I'm treated with respect, you know, I'm not treated like a child. They [Reintegration Centre staff] treat me with respect.* (P17)

In sum, the proximity of the Reintegration Centre to the detention centre promoted rapid access to reentry supports upon release. Participants identified a range of reentry priorities at intake, with housing being the most common. Peer support workers addressed some of these priorities immediately through the clothing, food, transportation, and harm reduction supports available on site. Together, the Reintegration Centre setting, services, and staff members not only helped reduce some of the immediate material hardship of reentry, but also affirmed participants' autonomy, agency, and worth as persons.

## Making referrals

When the Reintegration Centre could not immediately meet participants' reentry needs at intake, they attempted to address those priorities through referral to other programs within the JHST or other service agencies. At the time of Reintegration Centre intake needs assessment, most participants received referrals for several social support services (e.g., health treatment, housing programs, employment services, social benefits programs). At the time of the interview, most participants reported those referrals had been helpful. Connecting with the Reintegration Centre as a source of reentry support opened up opportunities for participants to learn about and connect with other supports.

As noted earlier, peer support workers asked participants about their priorities, such as housing, during the Reintegration Centre intake needs assessment process, which formed the

basis of referrals to JHST's suite of services or other service organizations. One participant noted, for example, that the Reintegration Centre not only made referrals to address his top three priorities, but also arranged his appointments with those agencies: *"[The peer support worker addressed] all three [priorities] actually . . . and she's the one who set the appointments too . . . Yeah, it was [very helpful]"* (P10).

Many referrals were to other programs within the JHST's suite of services, especially the employment and housing programs. For instance, during the follow-up study interview a client commented on the assistance of the JHST housing worker with whom he was connected, *"she's definitely been helpful you know sending me places that are for rent and that"* (P16).

Participants also talked about referrals to services outside JHST, for social assistance, mental health, employment and shelter services, highlighting the important feeder role of the Reintegration Centre in connecting this population with a variety of resources available in the community. For example, many participants, in addition to receiving harm reduction kits, were referred to drug treatment programs through the Reintegration Centre:

> *[Y]ou did the best you could in here I think. From my [initial] looking in, getting me back into society, getting me back into the treatment centres. Showing me a way to get back into help . . . If I didn't have this place [Reintegration Centre connecting me with treatment]. . . I'd probably be back on the streets, crack using, and back in drugs and jail again.* (P05)

Some participants, while appreciative of the opportunity for referral to other services, did not feel a need for them. For instance, one participant expressed drug treatment groups were not the right fit given his personality and patterns of use:

> *They [peer support workers] gave me a couple of options, but I wasn't sure I wanted to take those routes, like the counseling and shit—that's going to make me want it even more. Put me in a room full of people that are talking about the best times they had smoking dope [and] it's going to make me think, 'Hey, I'm going to go smoke.'* (P12)

Another participant noted he already had a reentry plan and pieces in place, so he did not need help from the Reintegration Centre:

> *No [I did not need more help from the Reintegration Centre] because, when I was here, I was just released that day, and I came back to get my belongings, and I was in the area, so I came here, and then I went to, I stayed at a friend's place and then I turned myself into [a drug and alcohol treatment program] on the 18th, and then I've been there ever since so. . .* (P20)

Relatedly, other clients adhered to a personal responsibility ethic when it came to their reintegration:

> *I don't need any help from the John Howard. I appreciate it though. . . I just need to stay out of trouble.* (P01)

Another thanked the Reintegration Centre for health referrals, but:

> *I've long been my own best doctor and advocate.* (P08)

In sum, the Reintegration Centre made referrals to programs within JHST and other service agencies when it could not immediately meet the particular needs of their clients. Most

participants received such referrals, and during the interview, most reported those referrals had been helpful. Indeed, at the time of the interview, two-thirds of priorities either had been met or were in progress. While the Reintegration Centre was not a panacea for all of the everyday challenges clients faced, the services received were helpful in assisting with the reentry process and providing a pathway to other supports. Relatedly, participant responses suggest that service utilization patterns were tethered to self-identified needs, strengths, and strategies.

### Becoming a trusted reentry resource

The Reintegration Centre experiences of most study participants engendered trust, empowerment, and a desire to maintain contact with the Reintegration Centre and JHST. The time and care Reintegration Centre staff members invested in participants set it apart as a source of support for people released from custody. These were significant commendations from those who know the system, know what is available, what is useful, and how they have been treated elsewhere:

> *I've used this as my main backbone, like this is my rock because no other agency in the last month has given me as much help as this place. I mean, that interview we had when I first got out, we sat for what two, three hours? No other agency would even give me the time of day. They wouldn't even give me a half hour . . .. every question that I've had, every venture that I want to try, every advice that I need, it's been answered. And it's very rare for me to find a program that does that.* (P07)

Peer support workers with experiences of criminal justice involvement were trusted reentry resources participants could count on to understand their struggles and offer useful strategies to overcome structural and individual challenges:

> *[At other organizations] I go to meet a worker, and you can tell that they don't have the life experience because they're just sitting there like a deer in headlights. No, you don't because you haven't walked a second in my shoes. You have, K [peer support worker–name withheld]. You have. That's why I like John Howard.* (P07)

For some participants, peer support workers represented hopeful futures for themselves, giving participants the sense that they too might make it, they too were employable:

> *Interviewer: I got hired. I got 47 convictions.*
>
> *Participant: That's the good thing. That's why I said John Howard is great. I mean, who hires people like that, like look at what he just said. He got hired, right?*
>
> *Interviewer: So there's hope.*
>
> *Participant: There is hope.* (P05)

Given the importance of the Reintegration Centre as a source of reentry support, a significant theme in participant responses was the desire for ongoing connection to and communication with JHST's staff and services. As one participant expressed, "*I still plan on being involved with John Howard. They've done a lot so far*" (P10).

For participants, the integrated suite of services JHST offered (e.g., the Reintegration Centre connects to JHST's housing program) also set it apart from other service agencies, for it provided not only ease of service access but also a trusted service source:

*It's all within, which is great because I'm the kind of person, I have severe trust issues. I don't want to be going to a million different workers and having to re-explain everything over and over and over again because I've had to do that for ten fucking years.* (P07)

Similarly, another participant said that he would like to stay in contact with the peer support workers because he trusted their advice and would be able to work with someone who knows him and his case well, avoiding case-explanation fatigue:

*I don't know if anyone's ever said this, but to be honest, just someone that I can trust to talk to . . . if I ever needed advice. . . because you already know what I've been through. You've been, I've been on your caseload for a month. . . you know what I've gone through. . . because I don't want to have to re-explain my situation to worker after worker.* (P07)

Initial Reintegration Centre encounters—could thus provide the foundation for enduring JHST relationships, working together to effect meaningful change in the lives of people who rarely receive respect or encounter opportunity:

*John Howard cares about people. . . They treat me with respect at the John Howard and I don't think there's not nothing they wouldn't do for me. My workers have gone beyond the call of duty . . . [A]ll I can say is that John Howard's been walking with me through these 11 months and even when I wasn't in the jail and there's nothing more that I needed but to be treated like a human being and know that I had a chance to stay sober with their support.* (P17)

In sum, for most participants, the Reintegration Centre was a trusted reentry resource. Participants pointed to some core, interrelated features of their Reintegration Centre experience that contributed to its trusted service status. First, participants noted that the amount of time and effort Reintegration Centre staff members put into their service encounters went above and beyond their typical experiences at other agencies. Second, the Reintegration Centre commitment to peer support enhanced not only the service experience, but also the credibility of the program among participants, who stressed the importance of working with staff members who had lived experience and expertise of their challenges. Third, the integrated suite of services available at the Reintegration Centre's parent organization, JHST, provided a foundation for consistent and familiar service encounters, reducing some of the burdens involved in accessing services across multiple organizations.

## Discussion

The purpose of this article was to report the main findings of a process evaluation of a novel reentry program—the Reintegration Centre—a service hub to support people immediately upon release from custody. Drawing on interviews with Reintegration Centre clients, the article described the Reintegration Centre model and service experience in the context of the complex challenges people face upon release.

Consistent with previous research, participants experienced a variety of reentry barriers and revealed the personal, social, organizational, and structural complexity of the delivery context [101–103]. Housing (i.e., homelessness, housing instability) was the most common, and in some ways, the most difficult challenge given the lack of affordable housing in the reentry environment, coupled with systemic poverty and under-employment. Many participants also contended with serious trauma, social isolation, mental health and substance use concerns. Structural stigma and discrimination based on a history of incarceration were additional

reentry barriers identified here and elsewhere [20,23,26]. This set of challenges make post-release services essential tools in the effort to improve reentry outcomes [16,17,21], for connecting to services at the time of release is critical to effectively address complex and multidimensional needs [104,105].

Located near a detention centre and staffed with peer support workers, the Reintegration Centre was designed to address people's basic needs and connect them to a continuum of services upon release from custody. For participants, the Reintegration Centre was a nexus of support. The area surrounding the detention centre had few services addressing the needs of the population. The Reintegration Centre's proximity to the detention centre was thus an important facilitator of service awareness, access, and utilization [106]. At the Reintegration Centre, participants could immediately access food and clothing supports as well as harm reduction resources. Participants also emphasized the role of the Reintegration Centre in increasing their awareness of other supports and resources available in the community, and facilitating connections with those sources if required. This was especially important for those with weak-to-nonexistent social support networks [25,60,62]. This integrated, coordinated service approach reduced some of the barriers and frustrations participants had come to expect when accessing services, such as the requisite retelling of "their story" with each individual service encounter, documenting the history and current state of the deficits, needs, and challenges for which they sought support.

Innovative models of service coordination and collaboration not only have the potential to reduce redundancy and increase efficiency across the service system but also can improve the effectiveness of individual interventions. Support systems for this population should ensure an efficient connection across programs and services to pool resources and increase service utilization through centralized access of services that promote empowerment [62].

Relatedly, and consistent with recent research [73], the peer support approach improved the interactional dynamics of the service encounter, the overall impressions of the Reintegration Centre, and even the trust in JHST as a whole. The marginalization and stigmatization experienced by this population coupled with a lack of informal and formal support structures make reentry a trying and emotionally-wrought experience. The compassion and concern peer support workers demonstrated in their interactions with participants contributed to participants' positive impressions of the Reintegration Centre and themselves. Peer support workers affirmed participants' autonomy and moral worth, fostering trust in the Reintegration Centre. Peer support workers facilitated a more seamless and person-centered service experience, acting as empathic, understanding liaisons for participants, connecting them to services and programs suited to their reentry needs, priorities, and situations. The peer support worker was, for some participants, the embodiment of hope, representing a future beyond the cycle of incarceration [107].

In terms of opportunities for improving program implementation, participants noted that better coordination and communication between the Reintegration Centre and the detention centre might have smoothed the transition from prison to release, a consideration for future iterations of the Reintegration Centre. This siloed approach where the criminal justice system and community services working independently of each other has been documented [108] with the importance of this specific interagency collaboration and coordination also reported in the literature [62,109–112]. Other research has demonstrated the value of pre-release peer support in the reentry process. For example, a systematic review found that pre-release peer support was a cost-effective model with evidence suggesting that peer-led interventions provided a sound source of support within the prison system and were effective in reducing risky-behaviors [113]. Beginning peer mentorship during custody and maintaining it post release was critical to determine client needs for community reentry according to an evaluation of a

peer mentoring case management driven reentry model [113]. Peer mentors would already have established connections with clients, meeting them upon release, with evidence suggesting that this facilitated client adherence to outpatient recovery, abstinence from alcohol and drugs, helping them to secure employment and housing and advance their education [114]. The Reintegration Centre, is uniquely placed to extend its services, leveraging the contacts made between clients and peer support workers. With these established connections in place, the Reintegration Centre's peer support workers, are well positioned to extend contact with clients on their reentry journeys, acting as mentors while providing knowledge and support as they help clients navigate a complex system of decentralized services. To this end, the Reintegration Centre could further extend its reach with the addition of onsite representatives of multi-sector human services, becoming a one-stop-shop service centre hub for direct and efficient access to services and supports at a critical time in the reintegration journey of clients. One evaluation found that a health service brokerage model (i.e., trained workers harnessing existing community services to ensure tailored supports for clients) in the early days of release increased health service utilization and improved health outcomes [81].

We also acknowledge the limitations of this process evaluation. First, considering the complexity of the needs among this population, and that the interviews were conducted within a short time after they were released from jail, the findings reflect a snapshot of their experiences in the early days after release and therefore do not offer a window into longer-term circumstances, relationships, processes, or outcomes. Still, research indicates that the first month after release is characterized by high health risks and poor health outcomes, such as high rates of overdose and mortality [35–38], which not only underscores the critical need for rapid supports, service connections, and continuity of care, but also warrants shorter interview follow-up periods when conducting health-informed reentry program evaluation research [35,37,115,116]. Participants emphasized that the support they received from the Reintegration Centre was a critical factor in helping them navigate the high-risk period immediately following release and begin addressing health concerns, participating in their local communities, and rebuilding their lives. These findings are consistent with what other studies have found: community-based, post-release support services and interventions play a critical and positive role in enhancing community reintegration and reducing the risk of reincarceration [21,117,118].

While the results shed light on the early reentry experiences of those participating in a novel intervention, another potential limitation of the study is that the findings draw on qualitative interviews with 21 men, a subset of the 209 individuals who were invited to participate in the research, which may have affected the range of perspectives we could solicit in this evaluation. The experiences and perspectives of interview participants may differ in substantive ways from others. More generally, we would also emphasize that the challenges of reentry—a situation in which individuals are juggling competing priorities and complex needs at a transitional time wrought with stress and hardship [25]—can also act as barriers to research participation [93]. For example, with respect to maintaining participant contact and scheduling interviews, the study of people facing personal, social and health challenges post-release often poses additional recruitment and retention challenges than researchers might encounter in general population or treatment samples. We experienced and adjusted to these challenges throughout the study, but there were 28 people who agreed to an interview, but later could not be reached for scheduling.

Future research could explore the needs and experiences of other formerly incarcerated persons and evaluate the Reintegration Centre model among specific population groups, such as the historically marginalized, for a potential wide scale rollout of the Reintegration Centre in other jurisdictions. For example, gender-, trauma-informed models of care might be required to adapt and implement the Reintegration Centre to support women who experience

incarceration, release, and reentry in distinct ways and contend with specific post-release challenges. Further, while the interviews for this evaluation typically occurred within one month of study enrolment, a critical post-release period, additional follow-up or longitudinal data would allow for a better understanding of mid- and longer-term reintegration successes and challenges especially in the context of access to and use of Reintegration Centre services. In this vein, the collection of outcome metrics would provide an ideal complement to thick description, helping to identify defined health and well-being outcomes.

## Conclusion

People released from incarceration have complex and intersecting needs, which require comprehensive and coordinated community-based support services. Thus, joint efforts across social systems (e.g., justice, health, employment, and education) are required. Partnerships between the public and private sectors can facilitate access to meaningful skill training programs, employment opportunities, and affordable and safe housing choices. Reentry approaches that link existing support services within the correctional system with the existing community-based support organizations/partners, can facilitate a more person-centered reintegration plan. This has the potential to enhance opportunities for successful reintegration of those released from correctional facilities. It might also contribute to reducing the risk of homelessness, health deterioration, victimization and exclusion, which in turn, can decrease recidivism and re-incarceration. Helping people who are released from custody to meet their personal and social needs [119] also reduces the direct and indirect economic, social, and psychological costs to the general public from increased housing instability and homelessness, unemployment, substance use disorders, emergency health care services, crime and criminal justice system involvement [37,120,121].

## Supporting information

**S1 File. Reintegration centre–needs identification tool.**
(PDF)

**S2 File. Qualitative interview guide.**
(PDF)

## Acknowledgments

The authors gratefully thank the clients for sharing their experiences. We thank the peer support workers for their significant contributions to data collection. We are grateful for the collaboration of John Howard Society of Toronto on this project and specifically Amber Kellen. In collaboration, Amber Kellen and Flora Matheson acquired the funding and designed and conducted the study. We also thank former St. Michael's Hospital staff (Alison Baxter, Parisa Dastoori, Natasha James, David Kryszajtys) who were involved in data collection and coding of the data for the study as a whole. The project was supported by the MAP Centre for Urban Health Solutions, St. Michael's Hospital and the Dalla Lana School of Public Health, University of Toronto.

## Author Contributions

**Conceptualization:** Arthur McLuhan, Tara Hahmann, Cilia Mejia-Lancheros, Sarah Hamilton-Wright, Flora I. Matheson.

**Data curation:** Sarah Hamilton-Wright, Guido Tacchini, Flora I. Matheson.

**Formal analysis:** Arthur McLuhan, Cilia Mejia-Lancheros, Sarah Hamilton-Wright, Guido Tacchini, Flora I. Matheson.

**Funding acquisition:** Sarah Hamilton-Wright, Flora I. Matheson.

**Methodology:** Sarah Hamilton-Wright, Flora I. Matheson.

**Project administration:** Sarah Hamilton-Wright, Guido Tacchini, Flora I. Matheson.

**Software:** Sarah Hamilton-Wright, Flora I. Matheson.

**Supervision:** Flora I. Matheson.

**Visualization:** Arthur McLuhan, Tara Hahmann, Cilia Mejia-Lancheros, Flora I. Matheson.

**Writing – original draft:** Arthur McLuhan, Tara Hahmann, Cilia Mejia-Lancheros, Flora I. Matheson.

**Writing – review & editing:** Arthur McLuhan, Tara Hahmann, Cilia Mejia-Lancheros, Sarah Hamilton-Wright, Guido Tacchini, Flora I. Matheson.

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
