## [Decision Letter · Decision Letter 0]

16 Mar 2022

PONE-D-22-01952Finding help and hope in a peer-led reentry service hub near a detention centre: A process evaluationPLOS ONE

Dear Dr. Matheson,

Thank you for submitting your manuscript to PLOS ONE. After careful consideration, we feel that it has merit but does not fully meet PLOS ONE’s publication criteria as it currently stands. Therefore, we invite you to submit a revised version of the manuscript that addresses the points raised during the review process. As you will see below, the reviewers were enthusiastic about the manuscript but felt that it needed some additional revision to meet the publication criteria. In particular, the framing and clarification requested by reviewer #1 will help readers to follow the evaluation that was conducted.

We look forward to receiving your revised manuscript.

Kind regards,

Andrea Knittel

Academic Editor

PLOS ONE

Journal Requirements:

[The project was funded by the Ontario Government and the Ontario Trillium Foundation [LP95159]. The project was supported by the MAP Centre for Urban Health Solutions, St. Michael’s Hospital and the Dalla Lana School of Public Health, University of Toronto.]

 [FM received project funding from the Ontario Government and the Ontario Trillium Foundation [LP95856]. The funders had no role in study design, data collection and analysis, decision to publish, or preparation of the manuscript. https://otf.ca/;
https://www.ontario.ca/]

4. Please note that supplementary tables should remain/ be uploaded as separate "supporting information" files.

Reviewers' comments:

Reviewer's Responses to Questions

**Comments to the Author**

1. Is the manuscript technically sound, and do the data support the conclusions?

Reviewer #1: Partly

Reviewer #2: Yes

2. Has the statistical analysis been performed appropriately and rigorously? 

Reviewer #1: Yes

Reviewer #2: N/A

3. Have the authors made all data underlying the findings in their manuscript fully available?

Reviewer #1: No

Reviewer #2: Yes

4. Is the manuscript presented in an intelligible fashion and written in standard English?

Reviewer #1: Yes

Reviewer #2: Yes

5. Review Comments to the Author

Reviewer #1: GENERAL COMMENTS

This article is a contribution to the literature on reentry filling a gap by providing qualitative findings that provide important information about the experience of justice-involved individuals in the reentry process. The paper could be improved with additional citations in the literature review to highlight the need for qualitative data on the lived experiences of individuals in the reentry process as well as citation to prior research that has employed a process evaluation research design utilizing qualitative data. The method section needs elaboration to explain the research design. Elaboration on the weaknesses and future research is needed in the discussion.

SPECIFIC COMMENTS

1. A section in the literature review is needed that discusses research on the lived experience of formerly incarcerated and the value of qualitative data in understanding the reentry experience beyond recidivism (e.g., Maruna’s (2001) Making good: How ex-convicts reform and rebuild their lives. American Psychological Association; Gunnison & Helfgott (2013) Offender reentry: Beyond crime and punishment. Lynne Rienner; Gunnison, E. & Helfgott, J.B. (2017). Critical Keys to Successful Offender Reentry: Getting a Handle on Substance Abuse and Mental Health Issues. The Qualitative Report, 22(8).

2. A section is needed in the literature reviewing prior process evaluation research that has utilized mixed methods and qualitative research design to examine program impact beyond recidivism (e.g., Helfgott, J.B., Gunnison, E., Collins, P., & Rice, S.K. (2018). The power of personal narratives in crime prevention and reentry: Process evaluation of the Seattle Police Department’s IF Project. Corrections Policy, Practice and Research, 3 (2), 1-24; Helfgott, J.B. & Gunnison, E. (2020). Gender-Responsive Reentry Services for Women Leaving Prison: The IF Project’s Seattle Women’s Reentry Initiative. Corrections: Policy, Practice and Research, 5(2), 65-88.).

3. A section on peer support programs for formerly incarcerated is needed in the introduction (for example, The IF Project https://www.theifproject.org/

4. The Method section should be titled Method (not Methods)

5. The Method section should include a subsection titled Participants describing the demographics of the participants. The results of the demographics of the participants included in the results section should be moved to the Method section.

6. A more thorough explanation of peer support workers is needed. This concept is introduced under the Recruitment subsection of the Method section. It is unclear until later in the paper that these peers are formerly incarcerated individuals. The last paragraph under data collection and instruments begins to explain this. This paragraph needs to come earlier after the discussion of participants in a procedure subsection.

7. Under Data Collection and Instruments in the Method section, additional detail is needed on the Reintegration Centre Intake Interview and the semi-structured qualitative interview guide. Both of these instruments should be included as appendices. Are these interview guides drawn from established risk/needs assessments such as the LSI-R? This is not clear. If not drawn from validated instruments, what is the rationale for the questions included and how were they developed, and what specifically are they designed to measure? This needs to be sufficiently explained.

8. In the data analysis subsection, citation to prior process evaluations and qualitative methodology is needed to support the use of the qualitative data analysis.

9. The demographic characteristics of participants noted in the results section should be moved to the method section under Participants. The additional details on the participants such as contacts and conflicts in the criminal justice system are suitably placed in the results section, but the demographics should be placed in the method section.

10. The results presented should be more specifically tied to the questions on the interview guide – questions should be restated in the results section.

11. On the last paragraph under Making referrals, the sentence, “For instance, during the follow-up….” Makes little sense. What mean is being reported here and why is it included in this paragraph. The sentence is confusingly placed.

12. The ns should be italicized in the results section.

13. In the discussion, Authors may want to note that the quote indicating that the participants appreciate the peer support and note they have not trusted staff at other agencies because they have not walked in their shoes is consistent with prior findings that show the impact of social distance between CCOs and formerly incarcerated and the benefits of peer support programs (e.g., Helfgott, J.B. & Gunnison, E. (2008). The influence of social distance on community corrections officer perceptions of offender reentry needs. Federal Probation, 72 (1), 2-12; Gunnison, E. & Helfgott, J.B. (2011). Factors that hinder reentry success: A view from community corrections officers. International Journal of Offender Therapy and Comparative Criminology, 55(2), 1-18; Helfgott, J.B. (1997). Exoffender needs versus community opportunity in Seattle, Washington. Federal Probation, 61, 12-24.

14. I am uncertain of the value of the table on participant characteristics and without explanation of the intake instrument used it is unclear how the items are helpful in measuring the impact of the program. Additionally, with the small n and the details provided, I have concerns about the anonymity of the participants. I recommend that if the authors keep the table, that a clearer explanation of the items included in relation to the process evaluation be discussed in addition to how the participant anonymity was insured.

15. The limitations of the study need to be elaborated on in the discussion. The biggest weakness is the small n and the use of qualitative data to measure program effectiveness. In addition there is very little discussion of how the qualitative interviews and the participant information provided constitutes a process evaluation. This needs to be discussed in more detail in both the introduction, the methodology, and the conclusion/limitations.

Reviewer #2: In this manuscript, the authors present a process evaluation of the John Howard Society’s Reintegration Centre based on data from qualitative interviews with 21 clients. The Centre provides peer support and connections to a variety of services (e.g., assistance with housing, transportation, employment) for individuals recently released from incarceration.

From the outset of the manuscript, the authors suggest that evaluations of reentry programs tend to focus on recidivism as the primary outcome and neglect other possible indicators of program success, such as clients securing housing, finding employment, and getting support for mental health and substance use problems. Although there is indeed a tendency to focus on recidivism as the primary outcome in reentry program evaluation research, there are still plenty of studies―with mixed results―that do look at intermediate outcomes such as employment, substance use, and the like (e.g., D'Amico & Kim, 2018; Lattimore et al., 2012). The authors should mention some of those programs and the extent to which they are effective in causing change in the types of outcomes the authors are concerned with in the current manuscript.

The authors are correct in asserting that former inmates face many practical challenges during the process of reentry and that, ideally, a program aimed at easing community reintegration should address these. However, I am skeptical about the extent to which a program that only addresses these issues in the short-term will actually be transformative for former inmates in the long run. Let’s say, for example, that the Reintegration Centre helps a client find a job after release. As noted by Petrich et al. (2022) in a recent review of theory and research on reentry programs, “employment alone will tend not to cause desistance if the offender continues to have problems such as low self-control, hostile attribution biases, and an inability to resist negative peer influences and s/he leaves work.” In other words, an effective program would need to pair services tackling practical needs with those targeting cognitive and behavioral needs. It seems as though the JHS’s Reintegration Centre does not really deal with the latter, and so I think the authors should engage with these issues at some point in the manuscript.

In the Method section of the manuscript, the authors note that “staff introduced the study to 209 clients, and 139 consented to be part of a study and to share their intake interview data with the research team, of which 49 also provided informed written consent to participate in a qualitative interview with the research team and 21 completed the interview.” I understand that getting former inmates to agree to participate in research can be a difficult task. However, 21 out of 209 participants is less than 10%. Is there any indication of the degree to which participants and non-participants were alike on relevant characteristics? As an example, prior research shows that attrition is higher in longitudinal studies among non-Whites participants and for persons with lower SES, IQ, self-control, and education (e.g., Claus et al., 2002; Fielding-Singh et al., 2019; Mullan Harris et al., 2019). Likewise, I would guess that the many of the program clients who elected not to participate in the current study would differ in important ways from those who were included and, thus, their experiences with the program may be substantively different. If you cannot determine how alike the two groups are, I would note this in the Limitations portion of the manuscript and discuss implications for your findings.

It seems as though interviews for this study were conducted by peer support workers – the very people charged with service delivery at the centre. The authors should address the possibility that participants’ responses during interviews may have been colored by social approval bias or otherwise. It is possible that participants would rate the centre more favorably when speaking to a worker at the centre than they might when talking to an uninvolved third party.

Works Cited

Claus, R. E., Kindleberger, L. R., & Dugan, K. C. (2002). Predictors of attrition in a longitudinal study of substance users. Journal of Psychoactive Drugs, 34, 69-74.

D’Amico, R., & Kim, H. (2018). Evaluation of seven Second Chance Act adult demonstration

programs: Impact Findings at 30 months. U.S. Department of Justice.

Fielding-Singh, P., Patel, M. L., King, A. C., & Gardner, C. D. (2019). Baseline psychosocial and demographic factors associated with study attrition and 12-month weight gain in the DIETFITS trial. Obesity, 27, 1997-2004.

Lattimore, P., Barrick, K., Cowell, A., Dawes, D., Steffey, D., Tueller, S., & Visher, C. A. (2012). Prison reentry services: What worked for SVORI evaluation participants? U.S. Department of Justice.

Mullan Harris, K., Tucker Halpern, C., Whitsel, E. A., Hussey, J. M., Killeya-Jones, L. A., Tabor, J., & Dean, S. C. (2019). Cohort profile: The National Longitudinal Study of Adolescent to Adult Health (Add Health). International Journal of Epidemiology, 48, 1415-1415k.

Petrich, D. M., Cullen, F. T., Lee, H., & Burton, A. L. (2022). Prisoner reentry programs. In E. L. Jeglic & C. Calkins (Eds.), Handbook of issues in criminal justice reform in the United States (pp. 335-363). New York: Springer.

6. PLOS authors have the option to publish the peer review history of their article (what does this mean?). If published, this will include your full peer review and any attached files.

Reviewer #1: No

Reviewer #2: No

---

## [Author Response · Author response to Decision Letter 0]

14 May 2022

EDITOR COMMENTS

COMMENT: the reviewers were enthusiastic about the manuscript but felt that it needed some additional revision to meet the publication criteria. In particular, the framing and clarification requested by reviewer #1 will help readers to follow the evaluation that was conducted.

RESPONSE: Thank you for your feedback and the opportunity to revise the paper. We made several substantive changes to the introduction, method section, and discussion based on Reviewer #1’s comments that sharpened the framing of the paper, better situating our case and contribution with respect to specific lines of inquiry and recent debates in reentry research. In particular, we added content on the lived experiences of individuals in the reentry process, that go beyond recidivism, engaging with some of the interactive, interpretive, processual, and contextual dimensions of reentry cited in the literature. We also added literature on peer support workers and cited specific studies that employed a process evaluation research design using qualitative data to identify the communities of inquiry that provided the epistemic background and emerging debates that informed the shape and direction of the paper, and as were among the primary audiences we intended to engage with our work. Similarly, to clarify the focus and intent of our process evaluation we more explicitly define our approach and detail the purpose of the study for reviewers and readers, while also locating our methodological and analytical choices within well-established conventions of the qualitative process evaluation tradition, in the revised manuscript and/or the responses to reviewers, as applicable and appropriate. Further clarification is provided on data collection and instruments used in the Method section. We detail the above noted changes and other requests for clarification and modification below in the comments to both Reviewers.

REVIEWER 1

GENERAL COMMENT: This article is a contribution to the literature on reentry filling a gap by providing qualitative findings that provide important information about the experience of justice-involved individuals in the reentry process. 

RESPONSE: Thank you for acknowledging the importance of this research.

GENERAL COMMENT: The paper could be improved with additional citations in the literature review to highlight the need for qualitative data on the lived experiences of individuals in the reentry process as well as citation to prior research that has employed a process evaluation research design utilizing qualitative data. The method section needs elaboration to explain the research design. Elaboration on the weaknesses and future research is needed in the discussion.

RESPONSE: We thank the reviewer for these suggestions. We considered and responded to them in our comments and revisions, the details of which are noted below in our responses to each of these specific concerns/areas.

COMMENT: A section in the literature review is needed that discusses research on the lived experience of formerly incarcerated and the value of qualitative data in understanding the reentry experience beyond recidivism (e.g., Maruna’s (2001) Making good: How ex-convicts reform and rebuild their lives. American Psychological Association; Gunnison & Helfgott (2013) Offender reentry: Beyond crime and punishment. Lynne Rienner; Gunnison, E. & Helfgott, J.B. (2017). Critical Keys to Successful Offender Reentry: Getting a Handle on Substance Abuse and Mental Health Issues. The Qualitative Report, 22(8).

RESPONSE: Thank you. We added detail on the lived experience of formerly incarcerated individuals and the value of qualitative data in understanding the reentry experience beyond recidivism in the Introduction. In addition to the helpful references the reviewer provided, we also cite related literature in that section as we engage some of the interpretive, interactive, processual, and contextual dimensions of reentry (e.g., release conditions, identities, emotions). 

COMMENT: A section is needed in the literature reviewing prior process evaluation research that has utilized mixed methods and qualitative research design to examine program impact beyond recidivism (e.g., Helfgott, J.B., Gunnison, E., Collins, P., & Rice, S.K. (2018). The power of personal narratives in crime prevention and reentry: Process evaluation of the Seattle Police Department’s IF Project. Corrections Policy, Practice and Research, 3 (2), 1-24; Helfgott, J.B. & Gunnison, E. (2020). Gender-Responsive Reentry Services for Women Leaving Prison: The IF Project’s Seattle Women’s Reentry Initiative. Corrections: Policy, Practice and Research, 5(2), 65-88.).

RESPONSE: Again, thank you for the helpful suggestion and references—we have reviewed and integrated the suggested citations to previous process evaluations that employed qualitative methods in the Introduction. Please note that our study provides the findings of a qualitative process evaluation, embedded in a larger study that also explored health using administrative medical records and shelter use (findings not yet published) and as such, to be true to the paper we did not emphasize the mixed-methods approaches to or aspects of process evaluation studies in the literature review. We note that this paper is embedded in the larger study in the first paragraph of the method section.

“This study is part of a larger participatory research (Green & Mercer, 2001), mixed-methods process evaluation (Khenti, Bobbili, & Sapag, 2019) of the Reintegration Centre, a reentry service hub for men released from custody.

To set the context, frame the approach, and establish the contribution of the study more clearly and fully for readers we clarified the purpose of the study and defined process evaluation in the introduction: 

“Process evaluation determines whether program activities have been implemented as intended and resulted in certain outputs. According to the Centers for Disease Control and Prevention, process evaluations can answer questions such as, ‘How well the program is working. The extent to which the program is being implemented as designed. Whether the program is accessible an (sic) acceptable to its target population (CDC, 2021: p2).’ Often process evaluations proceed with qualitative methods to assess client experiences with the intervention (Bess, King, & LeMaster, 2004)’.” 

Approaching process evaluations thusly, we clarify the basic analytic foci of qualitative process evaluations: identifying and describing program activities and outputs; assessing whether they were implemented as intended; and examining participants’ experiences of the program (both on the delivery and reception side). Consistent with established methodological principles and practices in the process evaluation tradition, the extent to which any study focuses on one or more of these concerns depends on a variety of factors, including the objectives of the program under study as well as the intended purpose, audience, and use of the evaluation. Still, while acknowledging flexibility in the design of process evaluations, to “examine of program impact” falls outside the scope of process evaluation. 

COMMENT: A section on peer support programs for formerly incarcerated is needed in the introduction (for example, The IF Project https://www.theifproject.org/)

RESPONSE: Thank you. We’ve added additional background to the introduction on peer support, and also moved related material previously located in the method section to the introduction, as the reviewer suggested. 

COMMENT: The Method section should be titled Method (not Methods)

RESPONSE: We have changed this as requested. 

COMMENT: The Method section should include a subsection titled Participants describing the demographics of the participants. The results of the demographics of the participants included in the results section should be moved to the Method section.

RESPONSE: This has been completed. The participant section is at the end of the Method section.

COMMENT: A more thorough explanation of peer support workers is needed. This concept is introduced under the Recruitment subsection of the Method section. It is unclear until later in the paper that these peers are formerly incarcerated individuals. The last paragraph under data collection and instruments begins to explain this. This paragraph needs to come earlier after the discussion of participants in a procedure subsection.

RESPONSE: Thank you. We state the following in the Method section (under the subheading The Reintegration Centre): 

“Staffed by peer support workers with lived expertise of criminal justice involvement…”. 

We acknowledge the need for more information on peer support workers, and added a procedure subsection to the Method section that provides further information on the peer support workers at the Reintegration Centre.

COMMENT: Under Data Collection and Instruments in the Method section, additional detail is needed on the Reintegration Centre Intake Interview and the semi-structured qualitative interview guide. Both of these instruments should be included as appendices. Are these interview guides drawn from established risk/needs assessments such as the LSI-R? This is not clear. If not drawn from validated instruments, what is the rationale for the questions included and how were they developed, and what specifically are they designed to measure? This needs to be sufficiently explained.

RESPONSE: The intake interview is an internal needs assessment that the John Howard Society Toronto developed with external consultation prior to the study. To our knowledge, this needs assessment is not based specifically on assessments such as LSI-R. For the purposes of sample description and to update this needs assessment for their internal purposes, we worked with John Howard staff to add questions of interest for this study and for the staff at the RC who wanted to develop a better understanding of priority needs. The following information has been added to the revised manuscript and validity/reliability study citations are included where applicable. Please note that we modified the terminology and now use “intake needs assessment”. We changed this terminology throughout the manuscript.

The following paragraphs on the intake needs assessment tool and interview guide have been added to/revised in the Method section:

“Quantitative data to describe the study sample was collected during the intake needs assessment interview between the client and the peer support worker. This tool is used at the RC to identify immediate needs of the clients and to set priorities for support in a short interview format. It includes questions on criminal history, ratings of health (Mawani & Gilmour, 2010; Meyer, Cifuentes, & Warren, 2011), parental roles, history of trauma (the latter adapted from the Brief Trauma Questionnaire) (Schnurr, Vielhauer, Weathers, & Findler, 1999), resiliency (Smith et al., 2008), gender and ethnic identity, social support, service access, head injury, immediate and priority needs, and client strengths. This needs assessment provides a snapshot of the key health and social needs the participants faced at immediate release from custody and provides insight into the context of program delivery and therefore for interpreting the qualitative findings of the process evaluation.” 

The qualitative interview guide was developed in collaboration between the research team and the RC staff to understand reentry experience, experiences with service delivery through the RC, experiences of reentry and reintegration in general and their priority needs at release. We provide additional details on the interview guide in the Method section under Data Collection and Instruments as noted in the paragraph below: 

“The interview guide was designed to provide an understanding of service delivery experiences through the RC including peer support, broader experiences of reentry/reintegration, and priority needs at release. Included in these broader areas of interest are topics such as resiliency, housing, health and illness, sources of personal support, coping strategies, family & parenting, relapse and harm reduction, finances and employment, education/training, and personal goals for the future.”

We have also added both the intake needs assessment and the interview guide as supplementary files as requested. 

COMMENT: In the data analysis subsection, citation to prior process evaluations and qualitative methodology is needed to support the use of the qualitative data analysis.

RESPONSE: Thank you. We added the following additional citations (Helfgott & Gunnison, 2020; Helfgott, Gunnison, Collins, & Rice, 2020; Patton, 2014). 

COMMENT: The demographic characteristics of participants noted in the results section should be moved to the method section under Participants. The additional details on the participants such as contacts and conflicts in the criminal justice system are suitably placed in the results section, but the demographics should be placed in the method section.

RESPONSE: We moved the demographic characteristics to the Method section as requested. 

COMMENT: The results presented should be more specifically tied to the questions on the interview guide – questions should be restated in the results section.

RESPONSE: Thank you for the suggestion, and while we agree with the general thrust and attendant methodological implication of the reviewer’s comment--i.e., stressing the linkage of interview questions and study findings--we elected not to restructure the presentation of the results in the manner suggested. 

Consistent with the conventions of qualitative data reporting in the social sciences and qualitative evaluation (Patton, 2014), especially those in the interpretive, interactionist tradition, the qualitative interview guide is not a static instrument to be repeatedly and mechanically applied in interview after interview, but rather is best envisioned and approached as an emergent, processual product—one that serves to orient research interviewers in the back and forth interaction of the interview encounter to the lines of questioning--including unstructured follow up questions and probes—that attend to the basic analytical issues of interest, which not only vary from study to study and project to project, but also evolve in the course of any particular study or project with foundations in the interpretive, interactionist tradition. Moreover, given the inherently contextual, multi-perspectival nature of human group life (at least these are basic tenets of the aforementioned qualitative research tradition with which we are most closely aligned), we recognize and have indeed witnessed the tendency of interviewees to surprise interviewers with the responses they give in relation to the questions we ask—i.e., one might ask about reentry program services and receive a response that bears more on reentry challenges more generally, and vice versa). In short, given these and other considerations beyond the scope of this response, we have maintained the structure and flow of the results, the presentation of which do align with the overarching aims of the evaluation.

Specifically, the purpose of this article is to report the findings from a qualitative process evaluation of the pilot program to describe the context of implementation (i.e., the context of community reentry as experienced by Reintegration Centre clients, particularly the reentry challenges that relate to the need for RC services) and the core program features/components (i.e., Reintegration Centre services); to assess the client experience of the program, including accessibility and acceptability, including the identification of barriers and facilitators to implementation (such as client barriers to access and the role of peer support workers), and push forward knowledge in this area not only to inform reentry policy, practice and research, but also, importantly, to inform any further development and refinement of the Reintegration Centre program, particularly its implementation plans and processes.

COMMENT: On the last paragraph under Making referrals, the sentence, “For instance, during the follow-up….” Makes little sense. What mean is being reported here and why is it included in this paragraph. The sentence is confusingly placed.

RESPONSE: We apologize for the confusion. We removed information on the mean and range from this sentence and added this detail to the last sentence of the recruitment subsection..

COMMENT: The ns should be italicized in the results section.

RESPONSE: Thank you. We italicized this in the table 1. 

COMMENT: In the discussion, Authors may want to note that the quote indicating that the participants appreciate the peer support and note they have not trusted staff at other agencies because they have not walked in their shoes is consistent with prior findings that show the impact of social distance between CCOs and formerly incarcerated and the benefits of peer support programs (e.g., Helfgott, J.B. & Gunnison, E. (2008). The influence of social distance on community corrections officer perceptions of offender reentry needs. Federal Probation, 72 (1), 2-12; Gunnison, E. & Helfgott, J.B. (2011). Factors that hinder reentry success: A view from community corrections officers. International Journal of Offender Therapy and Comparative Criminology, 55(2), 1-18; Helfgott, J.B. (1997). Exoffender needs versus community opportunity in Seattle, Washington. Federal Probation, 61, 12-24.

RESPONSE: Thank you kindly for providing several references that helped bolster our findings. We now cite these studies in the discussion and introduction sections of our manuscript. 

COMMENT: I am uncertain of the value of the table on participant characteristics and without explanation of the intake instrument used it is unclear how the items are helpful in measuring the impact of the program. Additionally, with the small n and the details provided, I have concerns about the anonymity of the participants. I recommend that if the authors keep the table, that a clearer explanation of the items included in relation to the process evaluation be discussed in addition to how the participant anonymity was insured.

RESPONSE: We have elected to retain table 1 but for clarity have re-termed the intake interview to the intake needs assessment. This tool is used at the Reintegration Centre to identify immediate needs of the clients and to set priorities for support in a short interview format. It includes questions on criminal history, ratings of health, parental roles, history of trauma, resiliency, gender and ethnic identity, social support, service access, head injury, immediate and priority needs, and client strengths. This needs assessment provides a snapshot of the key health and social needs the participants faced at immediate release from custody and provides the context for the qualitative findings. We note in the process evaluation provided (Process evaluation of the Seattle Police Department’s IF Project. Corrections Policy, Practice and Research, 3 (2), 1-24; Helfgott, J.B. & Gunnison, E. 2020) that these authors also provide a descriptive table of participants which when reading the paper we found useful to understand the context for that particular process evaluation. We thank the reviewer for providing this very helpful reference. 

With respect to participant anonymity we take extra precautions in our participatory research processes to ensure that participants with criminal history are protected. Our processes are vetted through our Research Ethics Board. We do not report Ns smaller than 5 when the counts reflect individuals. 

COMMENT: The limitations of the study need to be elaborated on in the discussion. The biggest weakness is the small n and the use of qualitative data to measure program effectiveness. In addition there is very little discussion of how the qualitative interviews and the participant information provided constitutes a process evaluation. This needs to be discussed in more detail in both the introduction, the methodology, and the conclusion/limitations.

RESPONSE: While we appreciate the concern for and consideration of sample size more generally, the potential limitations of small-n samples are linked to and inseparable from the nature of the group phenomenon under study, the specific aims of the study, as well as the established conventions of the field. Thus, while indeed small-n samples would limit claims regarding program outcomes, effects, and effectiveness--especially evaluations of well-established programs that have been implemented at scale (e.g., across multiple implementation sites) for sustained periods (e.g., enough time to implement the program, make necessary revisions and adjustments, and compare and assess multiple cohorts of clients)—small-n samples are much less of a limitation and thus a concern in the context of process evaluations of new programs in the early piloting phase of the implementation cycle, as was the case of our qualitative process evaluation of the pilot program implementation of the Reintegration Centre. The evaluation of programs in those early developmental days is fundamentally exploratory in nature and as such any attempt at determining program effectiveness would not only be premature but also fall outside the scope of the conventions of qualitative approaches to evaluating programs in such nascent stages of development. 

Consistent with approaches to qualitative research design in health research (See e.g., Vasileiou K, Barnett J, Thorpe S, Young T. (2018) who provide a systematic analysis of qualitative health research over a 15-year period with emphasis on journal requirements for sample size justification and examples from BMJ, BJHP and SHI. They report that sample sizes often vary based on principles of saturation or due to pragmatic considerations), the present study examined experiences of reentry and of the Reintegration Centre intervention for a sample of men at immediate release from a detention centre. The data provided in Table 1 and 2 speak to the complexity of health and social issues they faced e.g., no immediate access to housing or food. Reconnecting with the men was challenging and the complex nature of the reentry process makes reconnection even more difficult. While we hoped to have more participants—a common experience among qualitative researchers more generally--this did not materialize. Still, although we acknowledge that the perspectives were elicited from a relatively small sample, due to these constraints, with a statement added to this effect in the Discussion section, the data provided in the qualitative interviews were rich and detailed the context of reentry, immediate needs and prioritization of these needs. 

REVIEWER 2

COMMENT: From the outset of the manuscript, the authors suggest that evaluations of reentry programs tend to focus on recidivism as the primary outcome and neglect other possible indicators of program success, such as clients securing housing, finding employment, and getting support for mental health and substance use problems. Although there is indeed a tendency to focus on recidivism as the primary outcome in reentry program evaluation research, there are still plenty of studies―with mixed results―that do look at intermediate outcomes such as employment, substance use, and the like (e.g., D'Amico & Kim, 2018; Lattimore et al., 2012). The authors should mention some of those programs and the extent to which they are effective in causing change in the types of outcomes the authors are concerned with in the current manuscript.

RESPONSE: Thank you for providing these two citations. We incorporated these citations to acknowledge the varied outcomes, beyond recidivism, that have been explored in the program evaluation literature. Please note that our study provides the findings of a qualitative process evaluation and as such, to be true to the paper we did not include a section on program outcomes, although, as noted above, we cite some key sources that reference the varied outcomes that program evaluations have reported. 

COMMENT: The authors are correct in asserting that former inmates face many practical challenges during the process of reentry and that, ideally, a program aimed at easing community reintegration should address these. However, I am skeptical about the extent to which a program that only addresses these issues in the short-term will actually be transformative for former inmates in the long run. Let’s say, for example, that the Reintegration Centre helps a client find a job after release. As noted by Petrich et al. (2022) in a recent review of theory and research on reentry programs, “employment alone will tend not to cause desistance if the offender continues to have problems such as low self-control, hostile attribution biases, and an inability to resist negative peer influences and s/he leaves work.” In other words, an effective program would need to pair services tackling practical needs with those targeting cognitive and behavioral needs. It seems as though the JHS’s Reintegration Centre does not really deal with the latter, and so I think the authors should engage with these issues at some point in the manuscript.

RESPONSE: Thank you for this comment. While we understand the concern about the short-term nature of the intervention; indeed, it is a visit immediately after release from a nearby detention centre. That said, the time directly following release is a critical period in the reentry journey of the formerly incarcerated and can set the foundation for successful reintegration (LaVigne et al., 2008). The Reintegration Centre model is focused on serving clients’ immediate needs and offering assistance at an important transitional time. The Reintegration Centre, for example, does not directly help clients access housing, but the peers can provide warm referrals to shelters, John Howard Society housing programs, and housing programs at other partner agencies. They also provide referrals to mental health services, income benefit/employment, and substance use resources. As well, they provide immediate harm reduction education and access to naloxone given that overdose deaths (especially during the current crisis) accelerate just after release. They provide tokens for travel, Tim Horton’s gift cards, and snacks at the Reintegration Centre. Peers accompany clients to any/all service destinations when requested and offer to do so. The peer support program allows clients to feel empowered and hopeful by offering the chance to relate to another person who has successful achieved similar goals to the ones that they may have set for themselves and a “friendly hand” after the demoralization of custody(Fletcher & Batty, 2012; Sunderland & Mishkin, 2013). 

COMMENT: In the Method section of the manuscript, the authors note that “staff introduced the study to 209 clients, and 139 consented to be part of a study and to share their intake interview data with the research team, of which 49 also provided informed written consent to participate in a qualitative interview with the research team and 21 completed the interview.” I understand that getting former inmates to agree to participate in research can be a difficult task. However, 21 out of 209 participants is less than 10%. Is there any indication of the degree to which participants and non-participants were alike on relevant characteristics? As an example, prior research shows that attrition is higher in longitudinal studies among non-Whites participants and for persons with lower SES, IQ, self-control, and education (e.g., Claus et al., 2002; Fielding-Singh et al., 2019; Mullan Harris et al., 2019). Likewise, I would guess that the many of the program clients who elected not to participate in the current study would differ in important ways from those who were included and, thus, their experiences with the program may be substantively different. If you cannot determine how alike the two groups are, I would note this in the Limitations portion of the manuscript and discuss implications for your findings.

RESPONSE: We recruited a large group of RC clients to understand their health and criminal justice experiences using the intake data collected as part of the Reintegration Centre standard intake needs assessment form. Of 209 clients introduced to the study, 139 (67%) consented to share this information. Of the 139 participants who consented to this part of the project 49 agreed to the qualitative interview (35%). We would have welcomed additional participation for the qualitative interviews but recognize that potential participants were asked about the interview at the day of their release, a transitional time marked, for some, by stress and anxiety (Western, Braga, Hureau, & Sirois, 2016; Western, Braga, Davis, & Sirois, 2015), and when they were reaching out for immediate help from the Reintegration Centre staff. The environment around the Toronto South Detention Centre is a service desert and considerable distance from the city of Toronto. All of these circumstances – among other unknowns – may have influenced people’s interest in participating in the study. 

Thank you for your feedback. With respect to differences between individuals who participated in the qualitative interview and others who only consented to share needs assessment data, we added a sentence in the Discussion section that states that those who agreed to participate in the interview may have certain characteristics and life experiences that differ from the non-interviewed group, which could be reflected in responses regarding the Reintegration Centre. 

COMMENT: It seems as though interviews for this study were conducted by peer support workers – the very people charged with service delivery at the centre. The authors should address the possibility that participants’ responses during interviews may have been colored by social approval bias or otherwise. It is possible that participants would rate the centre more favorably when speaking to a worker at the centre than they might when talking to an uninvolved third party.

RESPONSE: Thank you for your comment. We acknowledge that participants might be encouraged to review the Reintegration Centre favorably for reasons including social approval or to ensure that they receive optimal support. While we cannot ensure participant’s responses are unbiased, the informed consent process did make clear that their care would not be compromised as a result of their feedback on the interview. As it pertains to peer support workers conducting interviews versus a researcher, there is research that suggests peer interviewers facilitate greater comfortability in the interview context and this typically results in improved research quality (Devotta, Pedersen, Woodhall-Melnik, & Matheson, 2020; Devotta et al., 2016). We now provide a few sentences in the discussion that make mention of the potential bias and benefits of peer interviewers.

---

## [Decision Letter · Decision Letter 1]

16 Aug 2022

PONE-D-22-01952R1Finding help and hope in a peer-led reentry service hub near a detention centre: A process evaluationPLOS ONE

Dear Dr. Matheson,

Thank you for submitting your manuscript to PLOS ONE. After careful consideration, we feel that it has merit but does not fully meet PLOS ONE’s publication criteria as it currently stands. Therefore, we invite you to submit a revised version of the manuscript that addresses the points raised during the review process.

We look forward to receiving your revised manuscript.

Kind regards,

Vanessa Carels

Staff Editor

PLOS ONE

Journal Requirements:

Reviewers' comments:

Reviewer's Responses to Questions

**Comments to the Author**

1. If the authors have adequately addressed your comments raised in a previous round of review and you feel that this manuscript is now acceptable for publication, you may indicate that here to bypass the “Comments to the Author” section, enter your conflict of interest statement in the “Confidential to Editor” section, and submit your "Accept" recommendation.

Reviewer #2: (No Response)

2. Is the manuscript technically sound, and do the data support the conclusions?

Reviewer #2: Yes

3. Has the statistical analysis been performed appropriately and rigorously? 

Reviewer #2: N/A

4. Have the authors made all data underlying the findings in their manuscript fully available?

Reviewer #2: No

5. Is the manuscript presented in an intelligible fashion and written in standard English?

Reviewer #2: Yes

6. Review Comments to the Author

Reviewer #2: After reading through the revised version of this manuscript, I appreciate several of the additions that the authors have made in an attempt to improve it. The discussions of prior qualitative research on reentry and desistance experiences generally helped to contextualize the current work more, and more information on the nature of qualitative process evaluations was helpful as well. Overall, I think that the manuscript is much-improved, but there are still a couple of lingering issues.

The in-text description of the JHS’s intake needs assessment (and its inclusion as an appendix) was also helpful, but for me this raises more questions that are unanswered in the current iteration of the manuscript. This study was intended to be a process evaluation, and a large part of the “process” of the Reintegration Centre revolves around referring clients to services on the basis of that needs assessment. The importance of “individualized service referrals” is mentioned specifically in the Abstract and elsewhere in the manuscript. However, the assessment does not appear to be “scored” in any actuarial sense, and it is not clear how peer support workers are trained to conduct this assessment with any sort of rigor. Can the authors provide any insight on that training and how support workers identify their clients’ greatest needs? Are there any ways that this assessment could be improved? Should the Centre adopt a validated, actuarial assessment tool like the LSI-R or otherwise in order to better support services in the areas clients need it the most?

In my review of the initial submission, I made note of the fact that only 21 people participated in this study out of an initial 209 who were approached. The authors have added a few sentences to the Discussion section noting this and the difficulties of recruitment, but these additions do not address the crux of the issue that I mentioned: People who choose not to participate in a study tend to differ in many ways from those who do participate. It is entirely possible that a large portion of people who chose not to participate in this study did not find the program to be useful, or that they differ in some other aspect germane to the findings of the study. This limitation should be noted substantively in the Discussion.

7. PLOS authors have the option to publish the peer review history of their article (what does this mean?). If published, this will include your full peer review and any attached files.

Reviewer #2: No

---

## [Author Response · Author response to Decision Letter 1]

29 Sep 2022

Reviewer #2:

Comment 1: After reading through the revised version of this manuscript, I appreciate several of the additions that the authors have made in an attempt to improve it. The discussions of prior qualitative research on reentry and desistance experiences generally helped to contextualize the current work more, and more information on the nature of qualitative process evaluations was helpful as well. Overall, I think that the manuscript is much-improved, but there are still a couple of lingering issues.

Response 1: Thank you.

Comment 2: The in-text description of the JHS’s intake needs assessment (and its inclusion as an appendix) was also helpful, but for me this raises more questions that are unanswered in the current iteration of the manuscript. This study was intended to be a process evaluation, and a large part of the “process” of the Reintegration Centre revolves around referring clients to services on the basis of that needs assessment. The importance of “individualized service referrals” is mentioned specifically in the Abstract and elsewhere in the manuscript. However, the assessment does not appear to be “scored” in any actuarial sense, and it is not clear how peer support workers are trained to conduct this assessment with any sort of rigor. Can the authors provide any insight on that training and how support workers identify their clients’ greatest needs? Are there any ways that this assessment could be improved? Should the Centre adopt a validated, actuarial assessment tool like the LSI-R or otherwise in order to better support services in the areas clients need it the most?

Response 2: 

The reviewer acknowledges that the details we added on JHS intake needs assessment were helpful but raised additional questions. 

Before responding to these concerns as carefully and comprehensively as we can, we preface what follows with what we see as the broader issue from which these concerns seem to spring.

That is, the thrust of the concerns hinge on a key distinction between JHS’s needs assessment as an interpretive, interactive, emergent process, on the one hand, and the RC intake form as an instrument that peers use to facilitate and inform that broader process (given the interactional affordances the object offers in situ). We delineate the distinction as well as the risks of conflation below, as we attempt to respond to the interrelated set of concerns raised by the reviewer.

We would emphasize that “referring clients to services” is not solely, or even mostly, “on the basis of that needs assessment [instrument—the intake form],” but rather on the basis of a broader interpretive and interactive process of assessment, comprising multiple, interrelated dimensions and subprocesses, including, for example: 

• Developing rapport

• Establishing trust

• Managing reservations

• Engaging in sympathetic introspection

• Identifying, defining, describing, and discussing strengths

• Identifying, defining, describing, and discussing challenges and obstacles

• Identifying, defining, describing, and discussing challenges and (service) needs 

• Taking stock of habits, dispositions, tendencies, preferences

• Identifying reentry priorities, establishing reentry goals, and making reentry plans 

• Considering options and alternatives

From this view, making effective referrals to services is contingent on not only developing a shared understanding (i.e., intersubjectivity) of the salient strengths, obstacles, opportunities, needs, and priorities at reentry, but also on establishing a social space and relational style that invites and fosters peer-client co-empowerment, with each exercising agency in shaping the referral process as it unfolds.

The responses to the intake form in the field were not scored “in any actuarial sense,” as the reviewer rightly notes, but this was by design. The context of clients heretofore is one in which their status as persons, their selves, their identities have been defined in significant ways by the metrics of imprisonment, risk assessment, discharge planning, and community reentry. When prospective clients arrived at the RC, they were coming from a context in which they continually confronted the ritual degradation of self via the people-processing institutions of the criminal justice system, often through ostensibly valid, objective instruments of quantitative assessment. Asking them to submit to yet another actuarial assessment may further marginalize prospective clients at the critical moment of release from custody, sabotaging peer-client rapport, and prompting refusals of referral.

The metricization of marginalization and the construction of risky subjects, to be sure, is also a feature of the program evaluation research literature more generally. And notwithstanding the importance of quantitative assessment, there are also risks associated with relying on quantitative instruments of assessment, such as reducing and oversimplifying complex processes to static indicators as well as reducing both the face and ecological validity of referrals. In short, there are tradeoffs associated with every approach, relative to others, one needs to ask what’s won? What’s lost? What’s a wash?

While the LSI-R is designed to identify people’s risks and needs with regard to recidivism, including their particular criminogenic needs, the goal of the John Howard Reintegration Center’s assessment tool was intended to facilitate a conversation around clients’ immediate needs to ensure appropriate referral to services and supports While the LSI-R is a widely used tool within the correctional system, we would not advise implementing the LSI-R in community based agencies that are designed to support and empower people, particularly at the moment of release. Notwithstanding that some have noted that risk/needs tools can be supplemented with more strengths-based assessment tools that empower, promote well-being, and have additional benefits such as providing invaluable case planning information [3, 4], the concern would be that the community based agencies would be seen as an arm of the correctional system, which and likely inhibit and erode any sense of security for clients.. 

Given the relational, strengths-based approach to assessment as a process to which the John Howard Reintegration Centre subscribed, issues such as training peer workers “to conduct this assessment with any sort of rigor” become less consequential given this more comprehensive view of assessment as a complex, contingent, patterned, yet not-fully-predictable process.

With respect to the training associated with the intake form, the formal training period of the peers predated the project, both in terms of its design and course. There’s also an important distinction between training and doing/implementing in practice, that we would highlight and emphasize. That is, although training would be within the scope of a process evaluation, much more consequential for our purposes is what people actually do in practice, including the formal and informal processes of “learning the ropes” along the way. And thus while we do provide some insight into the pedagogical approach to the formal training associated with the intake form, doing so risks misrepresenting the form, inflating its importance in the broader relational, emergent, interactive, interpretive process of needs assessment to which John Howard Society Toronto and the peers subscribed.

Relatedly, “[H]ow support workers identify their clients greatest needs” is multidimensional interpretive and interactive process, beginning with developing rapport, establishing trust, introducing the instrument, defining competencies and dispositions, puzzling through problems, pursuing solutions, clarifying concerns, engaging in ongoing reflection on opportunities and obstacles given current circumstances, and so on. As a subject of evaluation, training as an ongoing process, while analytically interesting and practically consequential, is beyond the focus and scope of this paper. We would, however, encourage future work on peer training as a social process.

The intake form was developed by/with the Centre for Addiction and Mental Health (CAMH: Toronto, Ontario) and the John Howard Reintegration Centre – and prior to the development of the research project – with several goals, including peer and client empowerment through questions that center on client well-being and strengths. With respect to the training and identification of client needs it is our understanding that, at the time of development, CAMH staff provided the peer support workers with training that would support the peers to complete the intake form using the principles of Motivational Interviewing, empowering the peer support workers with the freedom to complete the tool in a conversational manner. In essence the form is a 2-way empowerment tool for peer support workers and clients, moving away from deficit based models that dominate the field. In our opinion a strengths-based approach is essential to support people who have experienced the stigma of criminalization [1, 2]. – NOTE: text in green font is included in the revised manuscript.

Please note that we did not include additional information regarding this comment into the paper at this time. If the reviewer and editor feel some additional 

Comment 3: In my review of the initial submission, I made note of the fact that only 21 people participated in this study out of an initial 209 who were approached. The authors have added a few sentences to the Discussion section noting this and the difficulties of recruitment, but these additions do not address the crux of the issue that I mentioned: People who choose not to participate in a study tend to differ in many ways from those who do participate. It is entirely possible that a large portion of people who chose not to participate in this study did not find the program to be useful, or that they differ in some other aspect germane to the findings of the study. This limitation should be noted substantively in the Discussion.

Response 3: This limitation has been added to the paper as noted below.

“We do not know why 188 people approached to participate in the qualitative interview declined; they may differ in substantive ways from participants.”

1. Feingold ZR. The stigma of incarceration experience: A systematic review. Psychology, Public Policy, and Law. 2021;27(4):550.

2. Tyler ET, Brockmann B. Returning Home: Incarceration, Reentry, Stigma and the Perpetuation of Racial and Socioeconomic Health Inequity. Journal of Law, Medicine & Ethics. 2017;45(4):545-57. doi: 10.1177/1073110517750595.

3. Fedock G, Covington SS. Treatment of Incarcerated Women and Girls. 2022.

4. Wanamaker KA, Jones NJ, Brown SL. Strengths-based assessments for use with forensic populations: A critical review. International Journal of Forensic Mental Health. 2018;17(2):202-21.

---

## [Decision Letter · Decision Letter 2]

8 Dec 2022

PONE-D-22-01952R2Finding help and hope in a peer-led reentry service hub near a detention centre: A process evaluationPLOS ONE

Dear Dr. Matheson,

Thank you for submitting your manuscript to PLOS ONE. After careful consideration, we feel that it has merit but does not fully meet PLOS ONE’s publication criteria as it currently stands. Therefore, we invite you to submit a revised version of the manuscript that addresses the points raised during the review process.

Reviewer #3: I commend the authors for a very well written paper on an interesting and important study. I think they have addressed the previous reviewers' comments very well. From my own reading of the paper, I had very minor comments to raise which I list below:

-The sentence added in the limitations section ("We do not know why...") reads like it needs some context, I would encourage the authors to revise it.

-Under findings, page 23, the paragraph in the middle of the page "Interactions with social service providers..." needs a quote to support it.

-There are different font sizes in the paper (introduction vs findings).

- I would recommend that the authors proof-read the paper carefully as there are some minor editorial issues - e.g. line 303 - "One coder [AM] on those that addressed".

We look forward to receiving your revised manuscript.

Kind regards,

Yandisa Sikweyiya, PhD

Guest Editor

PLOS ONE

Journal Requirements:

Reviewers' comments:

Reviewer's Responses to Questions

**Comments to the Author**

1. If the authors have adequately addressed your comments raised in a previous round of review and you feel that this manuscript is now acceptable for publication, you may indicate that here to bypass the “Comments to the Author” section, enter your conflict of interest statement in the “Confidential to Editor” section, and submit your "Accept" recommendation.

Reviewer #2: All comments have been addressed

Reviewer #3: All comments have been addressed

2. Is the manuscript technically sound, and do the data support the conclusions?

Reviewer #2: Yes

Reviewer #3: Yes

3. Has the statistical analysis been performed appropriately and rigorously? 

Reviewer #2: N/A

Reviewer #3: N/A

4. Have the authors made all data underlying the findings in their manuscript fully available?

Reviewer #2: No

Reviewer #3: No

5. Is the manuscript presented in an intelligible fashion and written in standard English?

Reviewer #2: Yes

Reviewer #3: Yes

6. Review Comments to the Author

Reviewer #2: Thanks for the clarification on the processes involved in the intake assessments. I think that this is now addressed as well as possible in the manuscript.

Reviewer #3: I commend the authors for a very well written paper on an interesting and important study. I think they have addressed the previous reviewers' comments very well. From my own reading of the paper, I had very minor comments to raise which I list below:

-The sentence added in the limitations section ("We do not know why...") reads like it needs some context, I would encourage the authors to revise it.

-Under findings, page 23, the paragraph in the middle of the page "Interactions with social service providers..." needs a a quote to support it.

-There are different font sizes in the paper (introduction vs findings).

- I would recommend that the authors proof-read the paper carefully as there are some minor editorial issues - e.g. line 303 - "One coder [AM] on those that addressed".

7. PLOS authors have the option to publish the peer review history of their article (what does this mean?). If published, this will include your full peer review and any attached files.

Reviewer #2: No

Reviewer #3: No

---

## [Author Response · Author response to Decision Letter 2]

23 Dec 2022

Response to Reviewer #3

COMMENT 1: The sentence added in the limitations section ("We do not know why...") reads like it needs some context, I would encourage the authors to revise it.

RESPONSE: Good catch. The statement was added as per a previous reviewer’s suggestion. We agree, however, that it could be better phrased and contextualized. Thank you for the opportunity to do so. We have revised discussion of potential limitations as follows:

We also acknowledge the limitations of this process evaluation. First, considering the complexity of the needs among this population, and that the interviews were conducted within a short time after they were released from jail, the findings reflect a snapshot of their experiences in the early days after release and therefore do not offer a window into longer-term circumstances, relationships, processes, or outcomes. Still, research indicates that the first month after release is characterized by high health risks and poor health outcomes, such as high rates of overdose and mortality [35-38], which not only underscores the critical need for rapid supports, service connections, and continuity of care, but also warrants shorter interview follow-up periods when conducting health-informed reentry program evaluation research [35, 37, 115, 116]. Participants emphasized that the support they received from the Reintegration Centre was a critical factor in helping them navigate the high-risk period immediately following release and begin addressing health concerns, participating in their local communities, and rebuilding their lives. These findings are consistent with what other studies have found: community-based, post-release support services and interventions play a critical and positive role in enhancing community reintegration and reducing the risk of reincarceration [23, 117, 118]. 

While the results shed light on the early reentry experiences of those participating in a novel intervention, another potential limitation of the study is that the findings draw on qualitative interviews with 21 men, a subset of the 209 individuals who were invited to participate in the research, which may have affected the range of perspectives we could solicit in this evaluation. The experiences and perspectives of interview participants may differ in substantive ways from others. More generally, we would also emphasize that the challenges of reentry--a situation in which individuals are juggling competing priorities and complex needs at a transitional time wrought with stress and hardship[119]--can also act as barriers to research participation [93]. For example, with respect to maintaining participant contact and scheduling interviews, the study of people facing personal, social and health challenges post-release often poses additional recruitment and retention challenges than researchers might encounter in general population or treatment samples. We experienced and adjusted to these challenges throughout the study, but there were 28 people who agreed to an interview, but later could not be reached for scheduling. 

COMMENT 2: Under findings, page 23, the paragraph in the middle of the page "Interactions with social service providers..." needs a quote to support it.

RESPONSE: Thank you for the suggestion. In addition to the original citation to extant research supporting the statement (#100), we added illustrative material from our study data, as the reviewer suggested. The paragraph now reads as follows: 

Interactions with social service providers can often be frustrating, even humiliating, especially for people experiencing complex needs [100]. For example, when asked if there were times “when you reached out or needed help and you didn't get it?” one participant lamented, “All the time. That's regular. Nobody wants to help. Most people just want to act like… ‘Sit there and listen’-- and then tell you you're the problem, everything is your fault!” (P12). Others expressed similar frustrations. For instance, 

Well the [reentry] challenges are just trying to get everything done and nobody's helping me so I just, like I said, I just put my hat, panhandle and make my money to get by … I'm trying to get my income together and everything else just screwing me up … You know, I make an appointment … for the clothing bank and the old lady was a bitch with me on the phone … She wasn't paying attention. Oh, this is, rah, rah, rah, rah so yeah. I'm stressed out … [the disability support program] won't give me any money. Welfare won't give me any money. I have an appointment coming up [with another service provider] for housing, so if I don't have the income, I got no money. I got no income and they can't do anything for me so I'd lose my subsidized housing. It's just been rough [returning to the community] … All I do is put out my hat every day and panhandle and so I have enough money to live off of … and also running around trying to get my [disability support application] together, trying to do Welfare, trying to do whatever I can getting it together so I can get a life. (P13)

Similarly, after being informed that he might be in violation of his probation order -- almost immediately upon release -- for accessing the services of the Hub to arrange access to drug treatment, rather than reporting directly to a drug treatment program, another participant shared:

[I was told] ‘You might get charged for a breach, you know.’ I had enough, like when you have that pressure, it makes you want to go use … These people are … not even helping you and probation is supposed to help people, like you’re on probation, but they’re supposed to tell you, guide you, help you, and get you help. It seems like they don’t do that. (P05)

The reviewer and readers will also see that we provide and discuss additional examples of challenging, frustrating interactions in accessing services later on in the results—set in comparative context vis-à-vis Hub peer worker service encounters—in “Becoming a trusted reentry resource,” including, for example, the following responses from P07:

“I've used this as my main backbone, like this is my rock because no other agency in the last month has given me as much help as this place. I mean, that interview we had when I first got out, we sat for what two, three hours? No other agency would even give me the time of day. They wouldn't even give me a half hour …. every question that I’ve had, every venture that I want to try, every advice that I need, it’s been answered. And it’s very rare for me to find a program that does that.” (P07)

[At other organizations] I go to meet a worker, and you can tell that they don’t have the life experience because they’re just sitting there like a deer in headlights. No, you don’t because you haven’t walked a second in my shoes. You have, K [peer support worker – name withheld]. You have. That’s why I like John Howard.” (P07)

“I have severe trust issues. I don't want to be going to a million different workers and having to re-explain everything over and over and over again because I’ve had to do that for ten fucking years.... I don’t want to have to re-explain my situation to worker after worker.” (P07)

COMMENT 3: There are different font sizes in the paper (introduction vs findings).

RESPONSE: Thank you. This has been corrected. 

COMMENT 4: I would recommend that the authors proof-read the paper carefully as there are some minor editorial issues - e.g. line 303 - "One coder [AM] on those that addressed".

RESPONSE: Thank you for pointing out this typo, which we’ve addressed in the current revision. We also did a full proof of the paper and tracked changes.

1. Binswanger, I.A., et al., Release from prison--a high risk of death for former inmates. N Engl J Med, 2007. 356(2): p. 157-65.

2. Kouyoumdjian, F.G., et al., The health care utilization of people in prison and after prison release: A population-based cohort study in Ontario, Canada. PLoS One, 2018. 13(8): p. e0201592.

3. Zlodre, J. and S. Fazel, All-cause and external mortality in released prisoners: systematic review and meta-analysis. Am J Public Health, 2012. 102(12): p. e67-75.

4. Kouyoumdjian, F., et al., Health status of prisoners in Canada: Narrative review. Can Fam Physician, 2016. 62(3): p. 215-22.

5. Stewart, L.M., et al., Risk of death in prisoners after release from jail. Australian and New Zealand Journal of Public Health, 2004. 28: p. 32-36.

6. Verger, P., et al., High mortality rates among inmates during the year following their discharge from a French prison. Journal of Forensic Sciences, 2003. 48: p. 614-616.

7. Visher, C., N. LaVigne, and J. Travis, Returning home: Understanding the challenges of prisoner reentry. 2004, Urban Institute Justice Policy Center: Washington, DC. p. 1-240.

8. Marlowe, D.B., Evidence-based policies and practices for drug-involved offenders. The Prison Journal, 2011. 91(3_suppl): p. 27S-47S.

9. Williamson, M., Improving the health and social outcomes of people recently released from prisons in the UK. 2006, The Sainsbury Centre for Mental Health: London, UK.

10. Western, B., et al., Stress and hardship after prison. American Journal of Sociology, 2015. 120(5): p. 1512-1547.

11. Western, B., et al., Study retention as bias reduction in a hard-to-reach population. Proceedings of the National Academy of Sciences, 2016. 113(20): p. 5477-5485.

12. Guilcher, S.J., et al., "Talk with me": perspectives on services for men with problem gambling and housing instability. BMC Health Serv Res, 2016. 16(a): p. 340.

---

## [Editor Report · Decision Letter 3]

1 Feb 2023

Finding help and hope in a peer-led reentry service hub near a detention centre: A process evaluation

PONE-D-22-01952R3

Dear Dr. Flora I. Matheson,

We’re pleased to inform you that your manuscript has been judged scientifically suitable for publication and will be formally accepted for publication once it meets all outstanding technical requirements.

Kind regards,

Yandisa Msimelelo Sikweyiya, PhD

Guest Editor

PLOS ONE
---

## [Editor Report · Acceptance letter]

9 Feb 2023

PONE-D-22-01952R3 

Finding help and hope in a peer-led reentry service hub near a detention centre: A process evaluation 

Dear Dr. Matheson:

I'm pleased to inform you that your manuscript has been deemed suitable for publication in PLOS ONE. Congratulations! Your manuscript is now with our production department. 

Kind regards, 

on behalf of

Professor Yandisa Msimelelo Sikweyiya 

Guest Editor

PLOS ONE